

# Modelling of cup anemometry and dynamic overspeeding in average wind speed measurements

Troels Friis Pedersen[1], Jan-Åke Dahlberg[2]

[1]Department of Wind and Energy Systems, Denmark Technical University, Roskilde, 4000 Denmark
[2]Stockholm, 12944, Sweden

Correspondence: Troels Friis Pedersen (trpe@dtu.dk)

**Abstract**

Cup anemometers measure average wind speed in the atmosphere, and has been used for one and a half century by meteorologists. Within the last half century cup anemometers has been used extensively in wind energy to measure wind
resources and performance of wind turbines. Meteorologists researched on cup anemometer behaviour and found dynamic overspeeding to be of an inherent and significant systematic error. The wind energy community has strong accuracy requirements for power performance measurements on wind turbines and this lead in the last two decades to new research on cup anemometer characteristics, which was taken to a new level with development of improved calibration procedures, cup anemometer calculation models and classification methods.

Research projects in wind energy demonstrated by field and wind tunnel measurements, that angular response was a significant contributor to uncertainty, and that dynamic overspeeding was a significant but less important contributor. Earlier research was mainly made on cup anemometers with hemispherical cups on long arms, and dynamic overspeeding was considered an inherent and high uncertainty on cup anemometers. Newer research on conical cups on short arms showed that zero and low overspeeding at low to medium turbulence intensities is present. Different cup anemometer calculation models were
investigated in order to find derived overspeeding characteristics. The general and often used parabolic torque coefficient model show that zero overspeeding is present when the speed ratio roots of the torque coefficient curve go through the equilibrium speed ratio and zero. The two-cup drag model is a special case of the parabolic torque coefficient model, but with the second root being reciprocal to the equilibrium speed ratio. The drag model always results in a positive maximum overspeeding in the order of 1.1 times the turbulence intensity squared. A linear torque coefficient results in maximum
overspeeding levels equal to the turbulence intensity squared. Torque characteristics of a cup anemometer with hemispherical cups fits slightly well to the drag model, but a cup anemometer with conical cups do not fit to neither the drag model nor the parabolic model, but better to a partial linear model, and even better to an optimized torque model. Most accurate modelling of cup anemometer characteristics is at present made with the ACCUWIND model. This model uses tabulated torque coefficient and angular response data measured in wind tunnel. The ACCUWIND model is found in IEC wind turbine power
performance standards, where it is used in a classification system for estimation of operational uncertainties. For an actual comparison of two cup anemometers, with hemispherical and conical cups respectively, the influence of dynamic overspeeding was found to be relatively low compared to angular response, but for conical cups it was specifically low.

## 1   Introduction

Cup anemometry has since about 1980 been used intensively in the wind energy community to assess wind resources and to document wind turbine power curves. A strong trust to this simple instrument was due to a long history in meteorological measurements. A cup anemometer consists of a vertical shaft with cup-shaped "cups" mounted on the top to provide rotation in the wind. Robinson was known to be the first to develop a cup anemometer with four hemispherical cups, and to bring it into general use in 1846. The cup anemometer instrument was, however, suggested by Edgeworth many years before



(Cleveland 1888), (Waldo 1893). The last of the 19th century, meteorological offices researched on "the factor", the gain calibration value, with swirling machines. These machines were set to rotate with a long horizontal arm with a cup anemometer mounted at the outer end. The meteorologists investigated the influence of the cup radius to rotor arm radius in order to determine the "factor", the cup speed relative to the wind speed. Robinson made a theory on the factor, which deviated, however, from other investigations with swirling machines and with outdoor cup anemometer comparisons (Cleveland 1888),

(Waldo 1893). Recently, (Sanz-Andres 2014), investigated the factor with a number of different conical cup designs, cup sizes and cup arms. He presented a thorough overview of literature on cup anemometry divided into different categories of research in the same article. In the 20th century, many meteorologists studied the overspeeding effect, i.e. a tendency to measure a higher average wind speed in fluctuating wind. With minor effort they studied angular characteristics, i.e. the influence due to non-horizontal inflow angles. Despite the research made by meteorologists, the World Meteorological Organization has not

over time presented strong requirements on accuracy of wind speed measurements with cup anemometers in their reports. Since 1980 and up until 2014 the requirement was 0.5m/s below 5m/s and 10% above 5m/s, (WMO-8 2014). The wind energy community made from the start use of the research made by the meteorologists, but it was experienced that stronger requirements on measurement accuracy was needed. The community was forced to make its own research and development on cup anemometry.

The wind energy community started intensely to measure wind turbine power curves (relation between free wind speed and wind turbine power output) with cup anemometers in the early 1980`ies. European wind turbine test stations met regularly to discuss common issues, especially challenges with power curve measurements and certification of wind turbines. This led to a European Joint Wind Turbine Test Station Programme to develop common procedures. The European test stations used different types of cup anemometers for power curve measurements and wind resource measurements. They calibrated the

instruments in different types of wind tunnels with different procedures. An inter-calibration between the wind tunnels revealed in 1989 up to 11% difference in calibration of the same cup anemometer (Hunter 1989). The cooperation led to common requirements to use of cup anemometry and wind tunnel calibrations, (Hunter 1989b), and also to regular inter-calibrations. After years of improvement of procedures, harmonized and recognized measurements were set up in 1997 by MEASNET, a measurement organization implemented by the European test stations, (MEASNET 2023). MEASNET perform today regular

inter-calibrations with the goal of less than 0.5 % differences in calibrations between the participating wind tunnels. All calibration institutes are accredited, and are able to trace calibrations and uncertainties back to fundamental physical units.

Hunter and his colleague discovered also in 1989, (Barton 1989), different dynamic behaviour in step responses (sudden increase or decrease in wind speed). From step responses they determined the distance constant, defined as the distance the air flows past a cup anemometer during the time it takes the cup rotor to reach 63.2 % of the equilibrium speed after a step change

in wind speed. They used the Meteorology Office Handbook (HMSO 1981) as reference. The distance constant definition is is the same in the updated standard (ASTM 2017). They determined distance constants of five instruments used by the European test stations, among them a Risø P2445b cup anemometer with conical cups, and a Thies Classic cup anemometer with hemispherical cups. They reported distance constants for Risø of 2.8m and 2.1m for increasing and decreasing steps, respectively. For Thies they reported distance constants of 5.2m and 5.3m respectively. MacCready introduced the distance

constant concept already in 1965 (MacCready 1965), assuming that the distance constant was a fundamental instrument constant. With the step response measurements Hunter and colleague found evidence that the distance constant varied with the conditions, and not seemed to be a fundamental constant, but at the time, the implications were not studied further.

The recommendation on the use of cup anemometry (Hunter 1989b) was followed up ten years later by the IEA organization by improved recommended practices (IEA-11 1999), a document that was widely used in wind energy. However, the IEA

recommendation did not solve the problems of differences in power performance measurements, experienced in field measurements with different types of cup anemometers. The experienced differences in wind speed measurements lead to a European trade barrier between Germany and Denmark. Although accredited power curves were made in Denmark with Risø



cup anemometers, when exporting wind turbines to Germany, it was required that power curves were measured in Germany
with Thies cup anemometers. In 2001 Albers sketched up the situation (Albers et al. 2001), and described the dawning need

of a classification system for cup anemometer performance in order to consider operational uncertainties, and in the
SITEPARIDEN project he published differences in field comparisons between cup anemometers, among them the Risø and
Thies cup anemometers, (Albers 2001). A procedure for classification of cup anemometers was earlier proposed (Pedersen,
Paulsen 1997). The procedure made use of the two-cup drag model introduced by Schrenk (1929). The procedure was further
developed (Pedersen, Paulsen 1999) and classification of five commercial cup anemometers with the method were presented.

The definition of the preferred measured wind speed was also an issue. Not having a specific wind speed definition would
alone provide for an uncertainty of 1.9% at 20% turbulence intensity by the available wind speed definitions (horizontal,
vector, scalar), (Pedersen et al. 1996). An analysis of wind turbine output power in relation to the wind indicated, that the
10min vector-scalar averaged wind speed would be a suitable definition. The European CLASSCUP project, (Dahlberg et al.
2001) was initiated to develop an optimum vector-average design of a cup anemometer and to prepare a classification system

to allow users to select anemometer suited to specific requirements and to assess operational uncertainties. The result of the
CLASSCUP project was a cup anemometer design with a flat angular response, but unfortunately also with relatively high
overspeeding. A revised classification system was also proposed, using tabulated torque coefficient data in modelling, instead
of using fitted data to the drag or parabolic torque coefficient models. An example classification report was made on the Risø
cup anemometer, (Pedersen 2004).

Further studies of the wind speed definition, where flow inclination angles for both cup anemometers and wind turbine blades
were assessed, concluded that the most suitable definition of measured wind speed for power performance measurements was
the 10min average horizontal scalar wind speed (Pedersen 2004 ). For cup anemometry the horizontal wind speed definition
is also the most obvious, due to the vertical shaft and the cosine relationship to the tilted flow. The European ACCUWIND
project (Dahlberg et al. 2006) continued the research on cup anemometry with the horizontal wind speed definition. The

horizontal wind speed definition was confirmed to be preferable rather than the vector definition, (Eecen et al. 2006), largely
due to the fact that wind turbines also respond to inflow angles with a cosine function, but to a power of two to four. The
testing methods on cup anemometry were investigated for robustness, and the classification procedure was fine-tuned,
(Dahlberg et al. 2006). Five commercial cup anemometers were tested with the now so-called ACCUWIND classification
method, (Pedersen et al. 2006). Included in these tests was an improved Thies cup anemometer with conical cups (Thies First

Class).

The classification method was adopted in the IEC power performance measurement standard, (IEC-12-1 2005) in annexes I
and J. The classification system is a method to assess the operational uncertainties in field measurements, Class A for flat
terrain, Class B for complex terrain, and a Class S for arbitrary terrain and measurement conditions. Classes A and B are
appropriate for selection of cup anemometry for measurement campaigns, while the Class S is appropriate for uncertainty

assessment of specific measurement campaigns. The IEC standard provides methods to combine the operational uncertainty
of a cup anemometer with all other uncertainties with use of the uncertainty standard, later updated to (BIPM 2008). The IEC
classification method was continued in the revised standard, (IEC-12-1 2017), where cold climate classes C to D were added.
MEASNET institutes (members of the MEASNET organisation) provide accredited calibration as well as classification of cup
anemometers according to the IEC standard. The 2017 IEC standard was lately restructured into the power performance

standard, (IEC-12 2022), which reference the new measurement standard, (IEC-50-1 2022), to where cup anemometry was
transferred.

The classification method requires use of an appropriate cup anemometer model to simulate the systematic deviations when
taking the cup anemometer from the wind tunnel to the field. The cup anemometer model has to simulate field conditions, but
also the calibration conditions, from where traceability is transferred, and the model must fit well to the calibration constants.

The ACCUWIND model, the example model in the IEC standard, is a generic time domain model that simulates the response



of a 3D wind exposed to a cup anemometer, using tabulated data of torque, angular response and bearing friction. Simpler models with mathematical expressions for torque characteristics do not describe the characteristics of actual cup anemometers to a sufficient detail, and they are therefore less useful for classification purposes. The mathematical models, however, imply dynamic characteristics, which are useful in the assessment of the overspeeding effect. These models are investigated in detail
in the following chapters.

Two actual cup anemometer types are used to demonstrate the range of characteristics of cup anemometer types, found in both field comparisons and in laboratory tests, see Fig. 1. The types are the before mentioned Risø P2445b (similar to Risø P2546 in geometry and characteristics and only called Risø in the following) and the Thies 4.3303.22.000 (often called Thies Classic, and only called Thies in the following). The Risø cup anemometer has three conical 70mm diameter cups, and radius from
shaft centre to cup centre 58mm. The Thies cup anemometer has three hemispherical 79mm diameter cups, and radius from shaft centre to cup centre 120mm.

## 2    Verification of differences in characteristics of cup anemometers

In the European SITEPARIDEN project Albers (Albers 2001), where he compared several cup anemometers in an experimental field setup, he found systematically 3% lower values by the Risø cup anemometer, compared to the Thies cup
anemometer, see Fig. 1. The cup anemometers were calibrated in the same wind tunnel at the same time, so the differences were due to climatic influence parameters on the cup anemometers. Pedersen made another field comparison experiment (Pedersen, et al. 2002), and found significant systematic influence of turbulence between the Risø and Thies cup anemometers, see Fig. 2. The field experiments verified the high influence of especially turbulence on cup anemometer measurements. Papadopoulos made yet another field comparison project, (Papadopoulos 2001). Five cup anemometers were compared in
horizontal flow and also flow in tilted position. He observed up to 2% differences at 12% turbulence intensity, and concluded that differences could not be explained alone on distant constant values (ranging from 1.7m to 5m). The distance constant was in the IEA recommendation, (IEA-11 1999) considered an important parameter to indicate the influence of dynamic overspeeding.

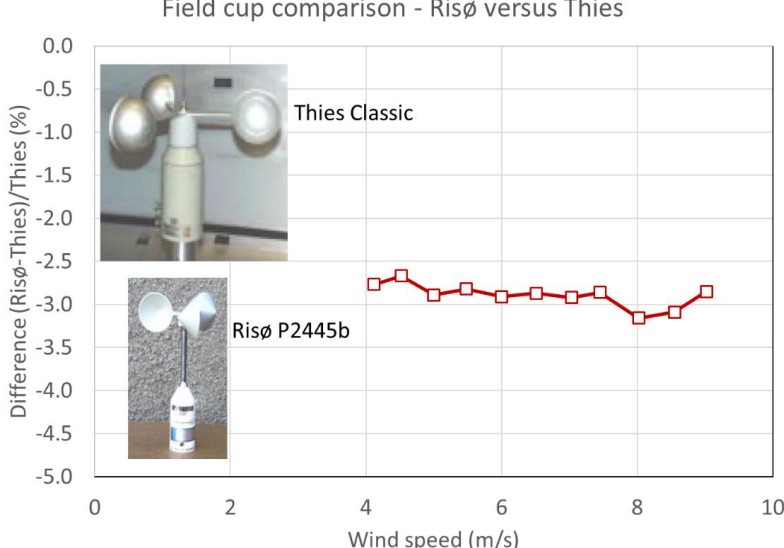


Figure 1 Differences of Risø cup anemometer relative to Thies in a field comparison at a measurement height of 8m, data from SITEPARIDEN, (Albers 2001)





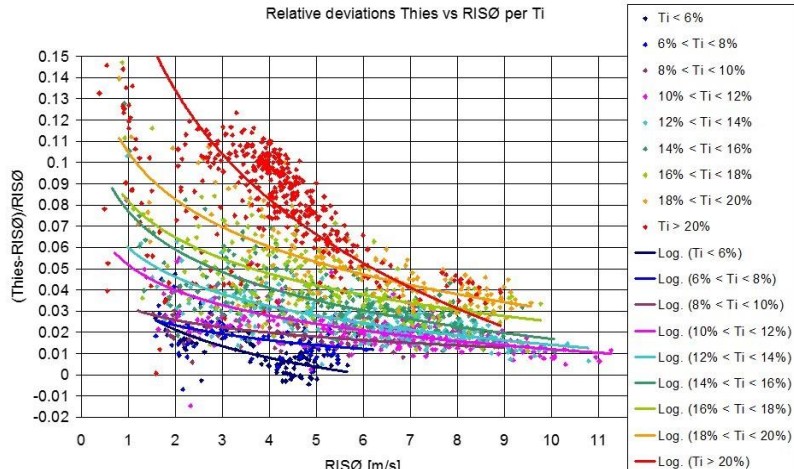

Figure 2  Differences of Thies cup anemometer relative to Riso as function of turbulence intensity, in a field comparison, from
(Pedersen et al. 2002)

## 3    Assessment of cup anemometer characteristics

By field experiments, the differences were discovered to be caused mainly by angular response, torque characteristics and
bearing friction. Bearings on cup anemometer rotors are lubricated with oil or grease, which is sensitive to temperature. At
lower temperatures, bearing friction increases and cup anemometers tend to measure lower wind speeds because of lower
rotational speed. Fabian (1995) demonstrated a fly-wheel test method in a climate chamber to assess bearing friction, and he
found significant friction differences between cup anemometer types. His method was adopted in the IEC standard, (IEC-12-
1 2005). Optimum bearing friction is obviously zero friction, or that friction is insensitive to temperature. Cup anemometers
are calibrated at indoor temperatures, but are used in field measurements, often at quite low temperatures. Temperature is thus
an important influential parameter, not to be neglected in field measurement uncertainty.

Turbulence is the most dominating influence parameter, as demonstrated in Figure 2. The turbulent wind gives rise to
instantaneous variations in upflow angle to the cup rotor. Wind tunnel calibration is made at horizontal flow, so the upflow
angle gives rise to an aerodynamic response due to non-horizontal angles. Angular response was studied in detail by
Westermann and Dahlberg in the CLASSCUP project (Dahlberg et al. 2001). Westermann made field tests of commercial cup
anemometers in tilted configurations, and Dahlberg made wind tunnel tests on commercial cup anemometers as well as several
potential designs for optimum flat angular response characteristics. Actual angular responses are generally not cosine shaped,
and they cannot easily be represented by mathematical formulas. Tabulated data was found to be most accurate when used for
cup anemometer classification. Several angular response characteristics for various cup anemometer configurations are
reported in (Dahlberg et al. 2001). Angular responses of the Thies and Risø cup anemometers are shown in Fig. 3.

Dynamic overspeeding is another turbulence effect due to horizontal wind speed variations. In static horizontal flow, the
equilibrium speed ratio (cup speed divided by wind speed) is determined by the calibration constants: gain and offset. In
turbulent wind, the rotor experiences off-equilibrium speed ratios due to wind variations and the cup rotor inertia, which causes
retardation of the rotational speed. The off-equilibrium rotor torque characteristics will then determine the amount of rotor
acceleration and deceleration, which causes the dynamical overspeeding effect.





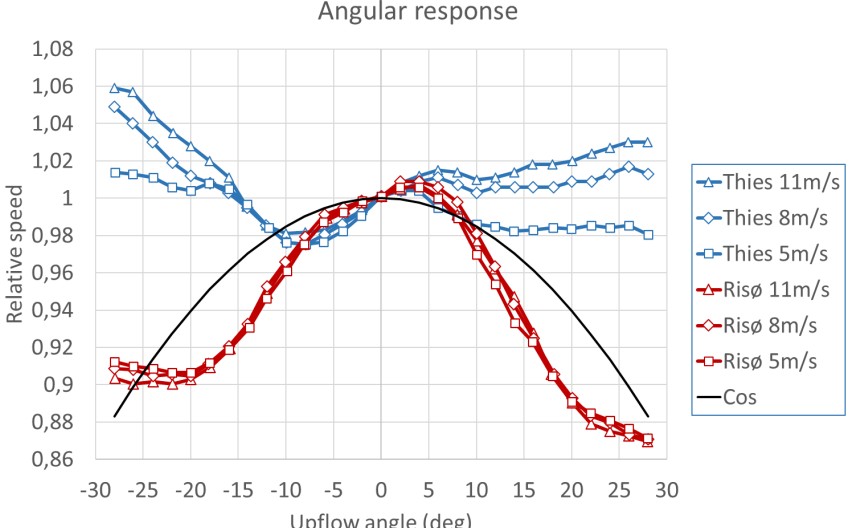

Figure 3 Angular response of Thies and Risø cup anemometers. Measurement data from (Dahlberg et al. 2001)

Schrenk (1929), in his pioneering work, made torque measurements on a hemispherical cup rotor at various tunnel wind speeds
and cup rotor rotational speeds. He normalized torque data to be generally dependent on the speed ratio alone. He fitted data
to a drag model, and also to a more general parabolic model, from which he calculated step responses and overspeeding effects.
Wyngaard (Wyngaard et al. 1974) made similar torque measurements, fitted these to a second order Taylor series expansion

perturbation model, and used it for simulating operation in the atmosphere. Busch and Kristensen, (1976), used the same
second order perturbation model in order to calculate overspeeding in the atmospheric boundary layer. Wyngaard later (1981)
considers the drag model, and makes a review of the research so far on cup anemometer dynamics. Coppin (1982) makes
torque measurements similar to Schrenk and Wyngaard on different types of cup anemometers, uses the second order
perturbation model, and finds significant differences between the cup anemometers. Dahlberg made torque measurements on

the Risø and Thies cup anemometers, shown in Fig. 4, (Dahlberg et al. 2006). Pedersen normalized Dahlbergs data with the
normalization procedure introduced by Schrenck, and found that it was valid in general. Schrenk (1929) was the first to
normalize wind tunnel torque measurement data into torque coefficient curves as function of speed ratio. He generalized the
torque coefficient with:

$$C_{QA}(\lambda) = \frac{Q_A}{\frac{1}{2}\rho ARU^2} \qquad (1)$$

Here $C_{QA}$ is the torque coefficient, $\lambda$ is the speed ratio, $Q_A$ is the rotor torque, $\varrho$ is the air density, A the projected area of one

cup, R the radius to the cup centre, and U the wind speed. He used the speed ratio:

$$\lambda = \frac{\omega R}{U} \qquad (2)$$

Where $\omega$ is the angular speed of the cup rotor.

However, Pedersen found the speed ratio definition not valid for stationary conditions, i.e. wind tunnel calibration conditions.
Calibrations have gain and offset. The bearing friction can explain some of the offset, but some offset remains due to
aerodynamic characteristics. This offset was named the "threshold wind speed", and was introduced into the expression of the

speed ratio in order to fit torque data to the calibration expression. The nature of the threshold wind speed has not yet been
explained:



$$\lambda = \frac{\omega R}{U - U_t} \quad (3)$$

The normalized rotor torque coefficient curves of the Risø and Thies cup anemometers from the ACCUWIND project are shown in Fig. 5.

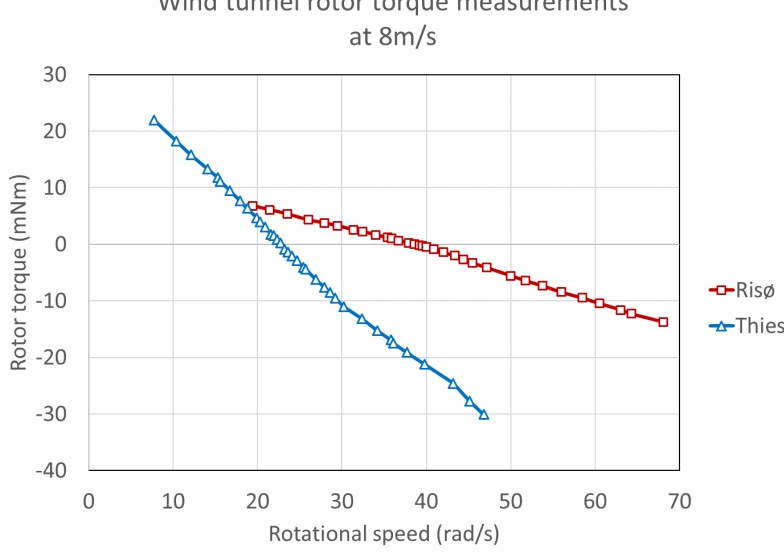

Figure 4  Rotor torque measurements of Risø and Thies cup anemometers in wind tunnel at 8m/s and varying rotor speed. Data from (Dahlberg et al. 2006)

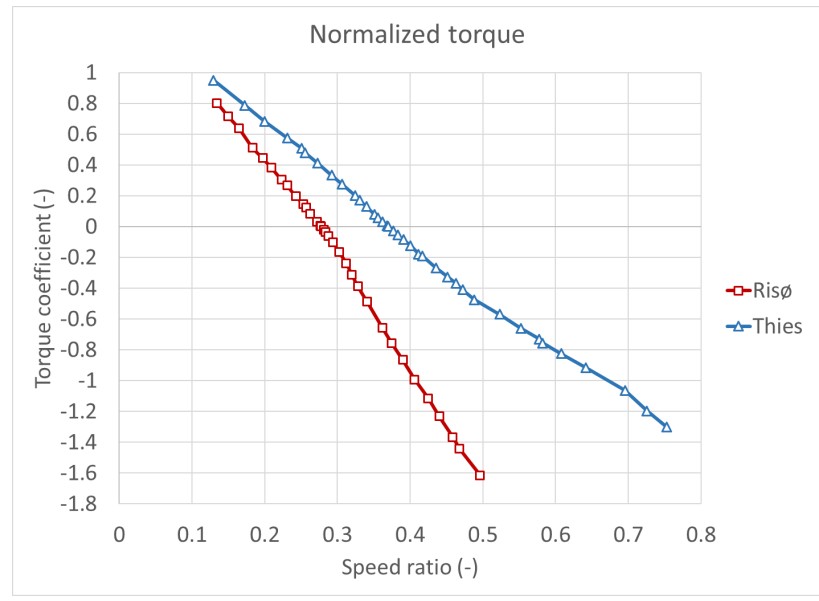

Figure 5 Rotor torque coefficient curves of Risø and Thies cup anemometers derived from Figure 4

In the CLASSCUP project Dahlberg also verified the influence of plane longitudinal wind variations. He generated sinusoidal air flow in the wind tunnel by rotating two vanes in the outlet of the tunnel, and he found directly the overspeeding of the Thies



and Risø cup anemometers at an average flow speed of 8 m/s, as shown in Fig. 6. The amplitude was varied by offsetting the angles between the vanes, and the frequency was varied by the rotational speed of the vanes. The Thies cup anemometer show significantly higher maximum overspeeding levels than the Risø cup anemometer. The overspeeding of the Risø cup
anemometer is slightly negative and close to zero at low turbulence intensity and low frequency.

Dahlberg (2006) also showed that torque measurements could be extracted from response tests with sinusoidal flow for very small time steps, assuming constant torque over one time step, using the formula:

$$C_{QA} = \frac{\Delta U}{\Delta t} \frac{2I}{\rho A R U^2} \tag{4}$$

He found good correlation between dynamic response tests and static torque measurements. He used dynamic response tests to make torque measurements of cup anemometers also in tilted conditions. He found that the overall response in tilted position
to a high degree was similar to the response when he first applied the influence of the angular response and afterwards the dynamic response at horizontal flow. The method by first applying the angular response and then the dynamic response was then considered robust, the procedure was adopted in the so-called ACCUWIND method.

The overspeeding curves in Figure 6 show clearly the overspeeding effects while the torque coefficient curves in Figure 5 reveal very little about the overspeeding effects. Most obvious from Figure 6 is the maximum overspeeding level at higher
frequencies, where the cup rotor inertia reduces rotational variations and keeps rotational speed practically constant.

The two cup anemometer types represent typical differences in overspeeding characteristics by cup rotors with hemispherical and conical cups, as investigated by Scrase (1944). He introduced general use of conical cups to the Met Office in London, in substitution of hemispherical cups. The differences in torque characteristics, and the advantage of conical cups were not discovered in wind energy before the CLASSCUP and ACCUWIND projects 2001-2006. The IEA document (IEA-11 1999)
did not mention this design difference.

The influence of dynamics by the cup rotor inertia is also evident in step responses, the response to a sudden change in wind speed. Maximum overspeeding and step responses describe the essence of dynamic overspeeding characteristics. They are therefore focus in the following assessment of cup anemometer models.

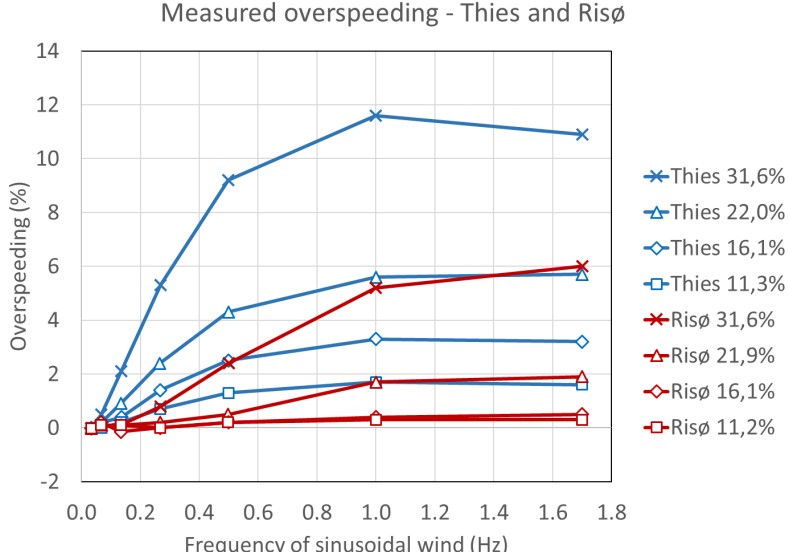

Figure 6  Dynamic overspeeding measurements in wind tunnel with sinusoidal wind speed variations. Average tunnel wind speed 8m/s and with turbulence intensities 11%, 16%, 22% and 32% (TI = $\Delta U/(U\sqrt{2})$). Data from ACCUWIND (Dahlberg et al. 2006)





## 4    Cup anemometer models

In comparison of different models, we use the same nomenclature for all models throughout the text. Models from historical references are only presented in this context if they consider integrated torque over one revolution.

All models start with an introduction of the general equation of dynamics that includes aerodynamic and bearing friction torque:

$$I\frac{d\omega}{dt} = Q_A - Q_F \tag{5}$$

Here I is the rotor inertia, $\omega$ is the cup rotor speed, $Q_A$ is the aerodynamic forces and $Q_F$ the frictional forces. $Q_A$ includes all aerodynamic forces due to wind from all directions. However, when dynamic effects are studied, the bearing friction torque is often omitted and only the horizontal unidirectional wind component is considered.

Schrenk presented a mathematical model of the cup anemometer, the two-cup drag model, see Fig. 7. On the left side, a high drag coefficient $C_{DH}$ represents the high drag due to the flow into the open cup while a low drag coefficient $C_{DL}$ represents the low drag due to the flow over the aerodynamically shaped front of the cup. The high drag on the left side will force the cup rotor to rotate clockwise while the low drag on the right side will reduce the clockwise rotation.

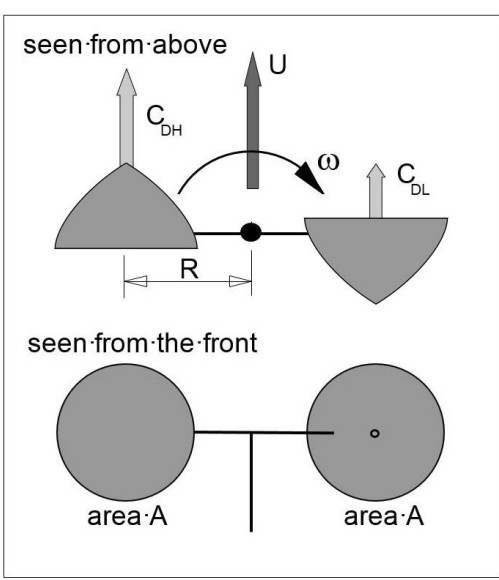

Figure 7  The drag model (not the original Schrenk sketch) with two cups on either side of the rotor. One cup on the left side with high drag coefficient $C_{DH}$ and another cup on the right side with low drag coefficient $C_{DL}$

The torque balance with the drag model is in this context expressed as:

$$Q_A = RD_H - RD_L = \frac{1}{2}\rho AR(U - R\omega)^2 C_{DH} - \frac{1}{2}\rho AR(U + R\omega)^2 C_{DL} \tag{6}$$

Here $D_H$ and $D_L$ are the drag corresponding to the two drag coefficients. The drag model is convenient because the low and high drag coefficients are the only constants containing aerodynamic properties to describe torque characteristics.

The drag model has since Schrenk been used by several authors, among others (Wyngaard 1981), (Westermann 1996), (IEA-11 1999), (Pedersen, Paulsen 1999), (Pindado et al. 2014). The drag model was widely considered a valid and simple model to describe the dynamics of cup anemometers.

Schrenk also presented a parabolic model with more constants. Here we present a fully flexible parabolic model with three convenient constants, including also Schrenks more general model. The three constants are descriptive and easy to understand.

 

They express the same as many other parabolic models, and lead to equivalent results. The torque coefficient is a parabolic function of the speed ratio $\lambda$, which has one specific root $\lambda_0$, the equilibrium speed ratio, related to the calibration expression. It is then obvious to add the second root of the parabola in formulation of the parabolic torque coefficient model. The model is then expressed as:

$$Q_A = \frac{1}{2}\rho ARU^2\beta(\lambda - \lambda_0)(\lambda - \lambda_1) = \frac{1}{2}\rho AR\beta(\omega R - \lambda_0 U)(\omega R - \lambda_1 U) \tag{7}$$

The first root, $\lambda_0 = \omega R/U$ relates to the gain of the calibration line $U = \omega R/\lambda_0$ (omitting the threshold wind speed and bearing friction). The second root $\lambda_1$ is a constant that basically determines the curvature of the torque coefficient curve in the area around the equilibrium speed ratio, and $\beta$ is an amplification factor that relates to the slope of the torque coefficient at equilibrium speed ratio:

$$\kappa = \beta(\lambda_1 - \lambda_0) \tag{8}$$

Second-order perturbation models include (Wyngaard et al. 1974), (Busch, Kristensen 1976) and (Coppin 1982). Rather than
considering the time domain, they considered second-order fluctuations or perturbations from equilibrium states. Kristensen (1998) mentions a phenomenological forcing model, based on more physical parameters. In principle, this model is also a parabolic torque model. The drag model, the perturbation models and the phenomenological forcing model all make use of second order or parabolic torque characteristics. Results obtained with these models will therefore all be similar to results obtained with the parabolic torque coefficient model presented in equation (7).

The linear model, with linear torque characteristics, even simpler than the parabolic model, is expressed as:

$$Q_A = \frac{1}{2}\rho ARU^2\beta(\lambda - \lambda_0) = \frac{1}{2}\rho AR\beta(\omega RU - \lambda_0 U^2) \tag{9}$$

The ACCUWIND model makes use of tabulated torque data, measured with a torque sensor in a wind tunnel, like Schrenk, Wyngaard and others. The normalization process uses the same expression for the torque coefficient, equation (1). However, the speed ratio is different from Schrenk's as Pedersen (Dahlberg et al. 2006) introduced the threshold wind speed $U_t$ in order to fit torque data to the calibration line, see equation 2.

In case the bearing friction is zero the calibration offset $B_{cal}$ reduces to the threshold wind speed $U_t$, and the expression for equilibrium speed ratio $\lambda_0$ is transformed into the linear calibration expression:

$$U = \frac{R}{\lambda_0}\omega + U_t = A_{cal}\omega + B_{cal} \tag{10}$$

When friction is applied, the calibration offset $B_{cal}$ gets larger than the threshold wind speed $U_t$ and the slope $A_{cal}$ increases slightly.

The ACCUWIND model is the most general cup anemometer model as it uses tabulated data for tilt response, normalized
torque and bearing friction. Normalization of the torque data is made by first extracting the friction torque from measured torque, then to normalize the aerodynamic torque with the target to fit $U_t$ to the calibration constants $A_{cal}$ and $B_{cal}$ at the calibration conditions. Fitting is made by simulation of the wind tunnel calibration, including the use of realistic turbulence intensity, for example 1% isotropic von Karman turbulence. A small speed ratio correction factor $\lambda_{corr}$ might be necessary to apply because torque measurements and calibration measurements may be made at different air temperatures and air densities
in the wind tunnel, and these differences are enough to disturb a correct fitting. Simulation with the ACCUWIND model is made with a 10min time-based 3D wind file, generated with the Mann model (1994, 1998). At each time step the instantaneous wind vector and upflow angle are determined. With the rotational speed and the upflow angle, the angular response is interpolated in the tabulated angular response data. The angular response is multiplied to the scalar of the wind vector to derive the resulting equivalent horizontal wind speed $U_{eq}$. The torque coefficient $C_{QA}(\lambda)$ is then derived by interpolation in the
normalized torque coefficient table with the $U_t$ adjusted speed ratio.





The friction torque $Q_F$ is found by interpolation in the friction table with the rotor speed ω and the air temperature T. Change in angular speed is found with the incremental time step Δt:

$$\Delta\omega = \frac{Q_A - Q_F}{I}\Delta t \qquad (11)$$

The actual response of a 10 min wind speed input with N time steps is determined by going through successive time steps to determine the "measured" wind speed $U = \sum_i(A_{cal}\omega_i + B_{cal})/N$. The "true" average of the horizontal input wind speed is

$U_{hor} = \sum_i(\sqrt{u_i^2 + v_i^2})/N$. The systematic deviation is determined by the "true" minus the "measured".

## 5   Overspeeding characteristics derived from cup anemometer models

Overspeeding characteristics of cup anemometers is best illustrated, as shown in Fig. 6, by the response to sinusoidal longitudinal horizontal wind variation and with step responses. The drag model, the parabolic model, the linear model, the partial linear model, and the ACCUWIND model are now assessed and compared for maximum overspeeding characteristics

and step responses.

### 5.1   Overspeeding with the ACCUWIND model

The overspeeding of the Risø and Thies cup anemometers, calculated with the ACCUWIND model, are shown in Fig. 8 and 9. The calculations show good agreement with the wind tunnel measurements, both with respect to the maximum overspeeding levels as well as the increase of overspeeding with frequency.

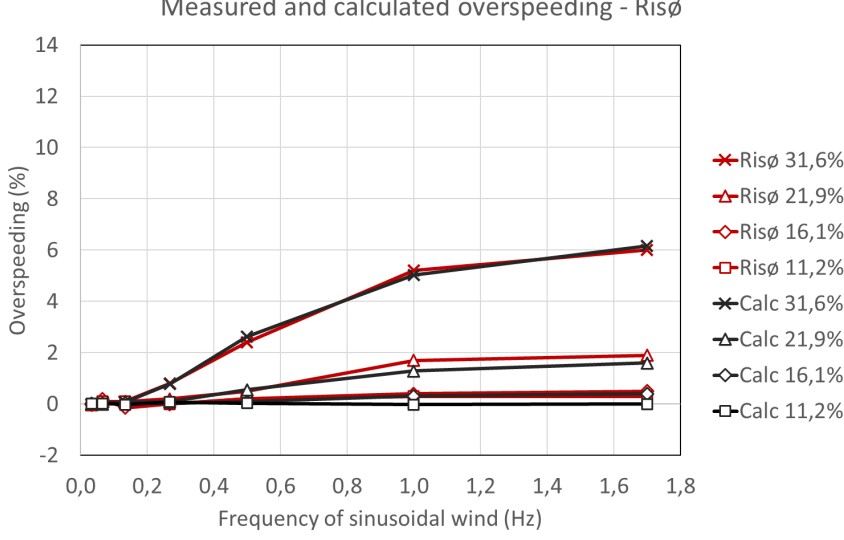


Figure 8   Dynamic overspeeding measurements and ACCUWIND calculations of Risø cup anemometer with sinusoidal wind speed variations. Average tunnel wind speed 8m/s and different turbulence intensities (TI = $\Delta U/(U\sqrt{2})$). Torque data from Figure 6



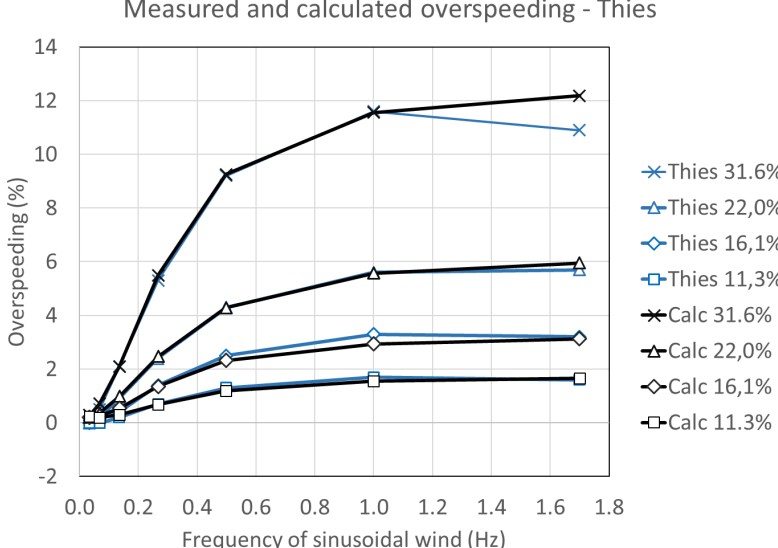

Figure 9 Dynamic overspeeding measurements and ACCUWIND calculations of Thies cup anemometer with sinusoidal wind speed variations. Average tunnel wind speed 8m/s and different turbulence intensities (TI = $\Delta U/(U\sqrt{2})$). Torque data from Fig. 6.

### 5.2 Maximum overspeeding with the parabolic torque coefficient model

The parabolic torque coefficient model is assessed for a typical equilibrium speed ratio $\lambda_0 = 0.3$ and for various values of $\lambda_1$,

see Fig. 10. The slope of the torque coefficient curves at the equilibrium speed ratio is set to $\kappa = -5$, which corresponds almost to the slope of the Risø torque coefficient curve in Fig. 5.

An expression of the maximum overspeeding level at high wind speed frequencies is derived from the parabolic torque coefficient expression, equation (7). Consider the cup anemometer being exposed to a sinusoidal wind speed $U_0 + \Delta U \sin(2\pi ft)$ at a sufficiently high frequency f where the cup rotor angular speed $\omega_0$ is constant due to the inertia of the cup

rotor. The instantaneous aerodynamic rotor torque is then:

$$Q_A = \frac{1}{2}\rho AR\beta(\omega_0 R - (U_0 + \Delta U \sin(2\pi ft))\lambda_0)(\omega_0 R - (U_0 + \Delta U \sin(2\pi ft))\lambda_1) \tag{12}$$



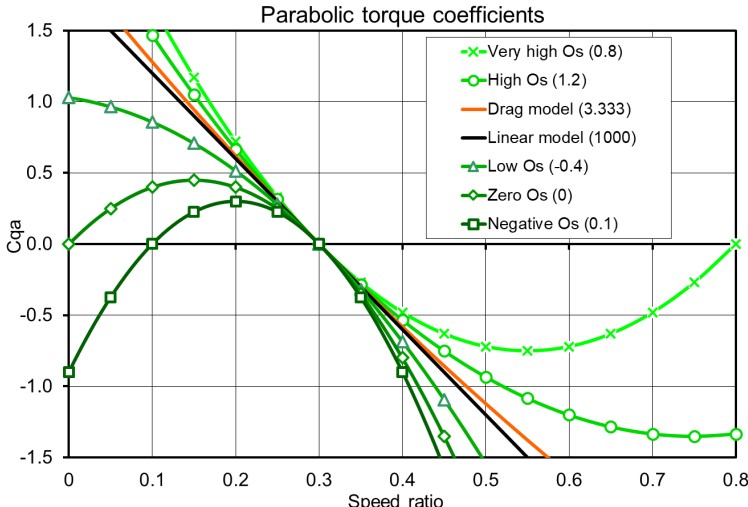

Figure 10  Parabolic torque coefficient curves for equilibrium speed ratio $\lambda_0 = 0.3$. Slope at the equilibrium speed ratio $\kappa = -5$ and various values of $\lambda_1$ as shown in the legend. Linear model in black and drag model in orange.


Now, integrating the torque over one cycle from $t = 0$ to $t = 1/f$ with the constant cup rotational speed $\omega_0$ we have:

$$\int_{t=0}^{t=1/f} Q_A dt = \frac{1}{2}\rho AR\beta (\omega_0^2 R^2 \int_0^{\frac{1}{f}} dt \tag{13}$$

$$- \omega_0 R(\lambda_1 + \lambda_0) \int_0^{\frac{1}{f}} (U_0 + \Delta U \sin(2\pi ft)) dt$$

$$+ \lambda_1 \lambda_0 \int_0^{\frac{1}{f}} (U_0^2 + \Delta U^2 \sin^2(2\pi ft) + 2U_0 \Delta U \sin(2\pi ft) dt\,)$$

Which integrates to:

$$\int_{t=0}^{t=1/f} Q_A dt = \frac{1}{2}\rho AR\beta \frac{1}{f} (\omega_0^2 R^2 - \omega_0 RU_0(\lambda_1 + \lambda_0) \tag{14}$$

$$+ \lambda_0 \lambda_1 \left(U_0^2 + \frac{1}{2}\Delta U^2\right))$$

Setting the integrated torque equal to zero we find the equilibrium angular speed $\omega_0$:

$$\omega_0 = \frac{U_0}{2R}(\lambda_1 + \lambda_0 \pm \sqrt{(\lambda_1 - \lambda_0)^2 - 2\lambda_0\lambda_1(\frac{\Delta U}{U_0})^2)} \tag{15}$$

We see that, setting the amplitude of the pulsating variations equal to zero, $\Delta U = 0$, we get the two roots:

$$\omega_0 = \frac{U_0\lambda_0}{R} \quad \wedge \quad \omega_0 = \frac{U_0\lambda_1}{R} \tag{16}$$

In case $\lambda_1 > \lambda_0$, the minus sign before the square root gives the first root, which is the equilibrium speed. In case $\lambda_1 < \lambda_0$ .the plus sign gives the first root.



The overspeeding is expressed as the angular speed increase in the pulsating wind divided by the angular speed in the constant
wind:

$$O_{s,max} = \frac{\omega_0 - \omega_{\Delta U=0}}{\omega_{\Delta U=0}} \tag{17}$$

$$= \frac{\frac{U_0}{2R}\left(\lambda_1 + \lambda_0 \pm \sqrt{(\lambda_1 - \lambda_0)^2 - 2\lambda_0\lambda_1(\frac{\Delta U}{U_0})^2}\right) - \frac{U_0\lambda_0}{R}}{\frac{U_0\lambda_0}{R}}$$

Which simplifies to:

$$O_{s,max} = \frac{1}{2}\left(\frac{\lambda_1}{\lambda_0} - 1 \pm \sqrt{\left(\frac{\lambda_1}{\lambda_0} - 1\right)^2 - 2\frac{\lambda_1}{\lambda_0}(\frac{\Delta U}{U_0})^2}\right) \tag{18}$$

The standard deviation of a sinus wave is the amplitude divided by the square root of two, so we have $\Delta U/U_0 = \sqrt{2}T_i$, where
$T_i$ is the turbulence intensity. The maximum overspeeding with a parabolic torque coefficient curve is then:

$$O_{s,max} = \frac{1}{2}\left(\frac{\lambda_1}{\lambda_0} - 1 \pm \sqrt{\left(\frac{\lambda_1}{\lambda_0} - 1\right)^2 - 4\frac{\lambda_1}{\lambda_0}T_i^2}\right) \tag{19}$$

The plus sign before the square root is used when $\lambda_1 < \lambda_0$ and minus when $\lambda_1 > \lambda_0$. Figure 11 shows the maximum
overspeeding of sinusoidal wind as function of turbulence intensity for the corresponding torque coefficient curves in Figure
10. The included maximum overspeeding values of the Thies are seen to be a little higher than the drag model, and are close
to follow the same pattern. The maximum overspeeding of the Risø, however, do not seem to follow neither of the curves, and
the parabolic torque coefficient model seem to fail completely in this case.

The expression in equation () is seen to depend only on the ratio of the roots $\lambda_1/\lambda_0$ and the turbulence intensity squared. From
the expression it is observed that the maximum overspeeding is zero when the second root $\lambda_1$ is equal to zero. This means
theoretically that dynamic overspeeding is fully eliminated when the torque coefficient curve is parabolic and the second root
is zero. The zero overspeeding is in this case independent of rotor inertia, distance constant, and frequency variations.

Kristensen (2002) made an analysis of overspeeding based on the "suspicion", discovered in the CLASSCUP project, that cup
anemometers might have zero or even negative overspeeding. He concluded that dynamic overspeeding is always positive,
while it can have negative overspeeding due to nonlinear calibration curves and angular characteristics below ideal
characteristics. The theoretical analysis shows, however, that dynamic overspeeding can actually be zero for parabolic torque
coefficients. Zero and even slightly negative overspeeding values are confirmed with the wind tunnel measurements on the
Risø cup anemometer at low turbulence intensities up to 16%,, while the overspeeding at higher turbulence intensities is
increasingly positive, (Dahlberg et al. 2001), (Dahlberg et al. 2006).


### 5.3   Maximum overspeeding with the drag model

An interesting case, also shown in Fig. 10 and 11, is the case of the drag model. Introducing the torque coefficient into the
drag model equation (6) and rearranging we get:

$$C_{QA} = \frac{Q_A}{\frac{1}{2}\rho ARU^2} = \left(1 - \frac{\omega R}{U}\right)^2 C_{DH} - \left(1 + \frac{\omega R}{U}\right)^2 C_{DL} \tag{20}$$

$$= (1-\lambda)^2 C_{DH} - (1+\lambda)^2 C_{DL}$$

Setting in the drag ratio $k = C_{DL}/C_{DH}$ we find the roots of the polynomial:



$$\lambda_0 = \frac{\sqrt{C_{DH}} - \sqrt{C_{DL}}}{\sqrt{C_{DH}} + \sqrt{C_{DL}}} = \frac{1 - \sqrt{k}}{1 + \sqrt{k}} \tag{21}$$

$$\lambda_1 = \frac{\sqrt{C_{DH}} + \sqrt{C_{DL}}}{\sqrt{C_{DH}} - \sqrt{C_{DL}}} = \frac{1 + \sqrt{k}}{1 - \sqrt{k}} = \frac{1}{\lambda_0} \tag{22}$$

We see that the drag model always has a second root reciprocal to the equilibrium speed ratio.

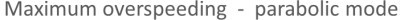

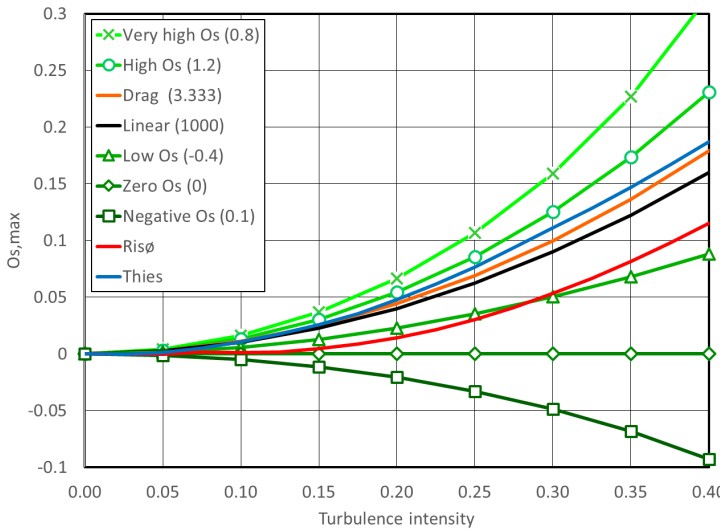

Figure 11 Maximum overspeeding of parabolic torque coefficient model for an equilibrium speed ratio $\lambda_0 = 0.3$ and various values of $\lambda_1$ as shown in the legend and as a function of turbulence intensity. Overspeeding of Risø and Thies are included.

The drag model is a special case of the parabolic torque coefficient model. The maximum overspeeding with the drag model is only dependent on the equilibrium speed ratio, and thus dependent on the slope $R/\lambda_0$ of the calibration line:

$$O_{s,max} = \frac{1}{2}\left(\frac{1}{\lambda_0^2} - 1 - \sqrt{\left(\frac{1}{\lambda_0^2} - 1\right)^2 - 4\frac{1}{\lambda_0^2}T_i^2}\right) \tag{23}$$

As the equilibrium speed ratio is dependent on the ratio between the drag coefficients, the maximum overspeeding is again dependent on the drag coefficient ratio:

$$O_{s,max} = \frac{2\sqrt{k} - \sqrt{4k - (1-k)^2 T_i^2}}{(1 - \sqrt{k})^2} \tag{24}$$

The maximum overspeeding for the drag coefficient model is always positive and a little higher than the turbulence intensity

squared. For a typical equilibrium speed ratio $\lambda_0 = 0.3$ the overspeeding is $1.1 \cdot T_i^2$, which for 10% turbulence intensity is 1.1% and for 20% turbulence intensity 4.4%. The drag model thus has a very specific torque coefficient curve and a very specific maximum overspeeding. The maximum overspeeding of the Thies cup anemometer in Figure 11 is 1.8%, to 5.8% for turbulence intensities 11%, to 22%. These maximum overspeeding values correspond to factors 1.5 to 1.2, which are somewhat larger than 1.10. The Thies cup anemometer is thus more prone to overspeeding than the drag model shows. Opposite with the

Risø cup anemometer, where the maximum overspeeding is 0.2% to 1.8% for turbulence intensities 11% to 22%. These maximum overspeeding values correspond to factors 0.2 to 0.4, which are much lower than 1.10. The drag model is thus





significantly overestimating the Risø cup anemometer overspeeding, while it underestimates the Thies cup anemometer. The cup shapes shown in Fig. 7 of the two-cup drag model are therefore not shown as conical cups, nor hemispherical cups, but something in between. Anyway, the drag model is representative for very limited types of cup anemometers, and is not

representative for modern conical cup shape anemometers being used in wind energy today. The parabolic torque coefficient model performs better because we can fit the data to each maximum overspeeding level at different turbulence intensities.

### 5.4 Maximum overspeeding with the linear torque coefficient model

Another interesting case, also seen in Figure 10 and 11, is the linear torque coefficient model with the torque expression:

$$C_{QA} = \beta(\lambda - \lambda_0) = \kappa(\lambda - \lambda_0) \tag{25}$$

In this case the slope at equilibrium speed ratio $\kappa$ is equal to the amplification factor $\beta$. With a sinusoidal wind and integrating

over one cycle the torque is:

$$
\begin{aligned}
\int_{t=0}^{t=1/f} Q_A dt &= \frac{1}{2}\rho AR\beta \int_{t=0}^{t=1/f}(U\omega_0 R - \lambda_0 U^2)dt \\
&= \frac{1}{2}\rho AR\beta \int_{t=0}^{t=1/f}(\omega_0 R(U_0 + \Delta U\sin(2\pi ft)) \\
&\quad - \lambda_0(U_0 + \Delta U\sin(2\pi ft)^2)(U\omega_0 R - \lambda_0 U^2)dt
\end{aligned} \tag{26}
$$

And resulting in the integral:

$$\int_{t=0}^{t=1/f} Q_A dt = \frac{1}{2}\rho AR\beta\frac{1}{f}(U_0\omega_0 R - \lambda_0 U_0^2 - \frac{1}{2}\lambda_0\Delta U^2) \tag{27}$$

Setting the torque equal to zero we find the equilibrium speed:

$$\omega_0 = \frac{U_0\lambda_0}{R} + \frac{1}{2}\frac{\lambda_0\Delta U^2}{RU_0} \tag{28}$$

And the overspeeding of a linear torque is:

$$O_{s,max} = \frac{\frac{U_0\lambda_0}{R} + \frac{1}{2}\frac{\lambda_0\Delta U^2}{RU_0} - \frac{\lambda_0 U_0}{R}}{\frac{\lambda_0 U_0}{R}} = \frac{1}{2}\frac{\Delta U^2}{U_0^2} = Ti^2 \tag{29}$$

A linear torque coefficient may also be achieved from the parabolic torque coefficient model when $\lambda_1$ is going towards $\infty$ or $-\infty$. In both cases we find that the maximum overspeeding for a linear torque coefficient is directly proportional to the turbulence intensity squared $O_{s,max} = T_i^2$. This is illustrated in Fig. 10 with the curve for $\lambda_1 = 1000$. This is about 10% less than the maximum overspeeding of the drag model. The linear torque model is thus not either able to model the torque characteristics of the Thies and Risø cup anemometers to a satisfactory level for the same reasons as for the drag model.

### 5.5 Maximum overspeeding with the partial linear torque coefficient model

Of more interest is the partial linear torque coefficient model with two linear torque coefficient curves, one at either side of the equilibrium speed ratio. The partial linear torque coefficient model is useful if a cup anemometer torque coefficient curve with an approximation can be considered partial linear in a broad range around the equilibrium speed ratio. The partial linear torque coefficient model was investigated by Pedersen (2011). He found that with the torque in this model he could achieve

almost the same results in classification of five types of cup anemometers as with tabulated data in the ACCUWIND model. The partial linear torque coefficient curves may be expressed as:





$$\text{if } \lambda \leq \lambda_0 : C_{QA} = \kappa_{low}(\lambda - \lambda_0) \tag{30}$$
$$\text{if } \lambda > \lambda_0 : C_{QA} = \kappa_{high}(\lambda - \lambda_0)$$

For the partial linear torque coefficient model the maximum overspeeding level can be determined by applying a sinusoidal wind speed as for the linear torque coefficient model. Consider again the cup anemometer to be exposed to a sinusoidal wind speed $U_0 + \Delta U \sin(2\pi f t)$ at a sufficiently high frequency f where the rotor angular speed can be assumed constant at $\omega_0$.

Now, integrating again the torque over one cycle from $t = 0$ to $t = 1/f$ with constant speed ratio $\omega_0$ we add the torque on each side:

$$\int_{t=0}^{t=1/f} Q_A dt \cong \int_{t=0}^{t=1/2f} Q_{A,low} dt + \int_{t=0}^{t=1/2f} Q_{A,high} dt \tag{31}$$

The approximation sign is due to the fact that the torque on either side is not exactly half of each cycle, but this is an error that

is very small and omitted here. Using the results from the linear torque coefficient model and setting the integrated torque equal to zero we find the equilibrium angular speed $\omega_0$:

$$\omega_0 = \frac{U_0 \lambda_0}{R} \cdot \frac{1 + \frac{4\Delta U}{\pi U_0} \frac{\kappa_{low} - \kappa_{high}}{\kappa_{low} + \kappa_{high}} + \frac{\Delta U^2}{2U_0^2}}{1 + \frac{2\Delta U}{\pi U_0} \frac{\kappa_{low} - \kappa_{high}}{\kappa_{low} + \kappa_{high}}} \tag{32}$$

The maximum overspeeding is thus:

$$O_{s,max} = \frac{\omega_0 - \omega_{\Delta U=0}}{\omega_{\Delta U=0}} = \frac{\frac{\Delta U^2}{2U_0^2} + \frac{2\Delta U}{\pi U_0} \frac{\kappa_{low} - \kappa_{high}}{\kappa_{low} + \kappa_{high}}}{1 + \frac{2\Delta U}{\pi U_0} \frac{\kappa_{low} - \kappa_{high}}{\kappa_{low} + \kappa_{high}}} \tag{33}$$

As $\Delta U / U_0 = \sqrt{2} \cdot T_i$ the expression is converted to:

$$O_{s,max} = \frac{T_i^2 + \frac{2\sqrt{2}}{\pi} \cdot \frac{\kappa_{low}/\kappa_{high} - 1}{\kappa_{low}/\kappa_{high} + 1} T_i}{1 + \frac{2\sqrt{2}}{\pi} \cdot \frac{\kappa_{low}/\kappa_{high} - 1}{\kappa_{low}/\kappa_{high} + 1} T_i} \tag{34}$$

When $\kappa_{low} = \kappa_{high}$ we get $O_{s,max} = T_i^2$ as for the full linear torque coefficient curve. Partial linear torque coefficient curves are shown in Fig. 12 for various $\kappa = \kappa_{low}/\kappa_{high}$ ratios. The maximum overspeeding of cup anemometers with partial linear torque coefficient is shown in Fig. 13. The maximum overspeeding of the Thies is seen almost to follow the ratio 1.2 curve, and the shape is quite similar. The maximum overspeeding of the Risø seem to follow close to the ratio 0.8 curve. This indicates that the partial linear model seem to be a better fit to the two cup anemometers than the parabolic torque coefficient model. It

confirms the experience that the partial linear model performs quite well in classification of the cup anemometers (Pedersen 2011).

We cannot achieve maximum overspeeding equal to zero for all turbulence intensities as for the parabolic torque coefficient model when $\lambda_1 = 0$. We have zero maximum overspeeding for the following $\kappa$ ratios:

$$\frac{\kappa_{low}}{\kappa_{high}} = \frac{4 - \pi\sqrt{2}T_i}{4 + \pi\sqrt{2}T_i} \tag{35}$$

For turbulence intensities 5%, 10%, and 15% the optimum $\kappa$ ratios are, for example 0.89, 0.80 and 0.71,

respectively. Linear model in black.



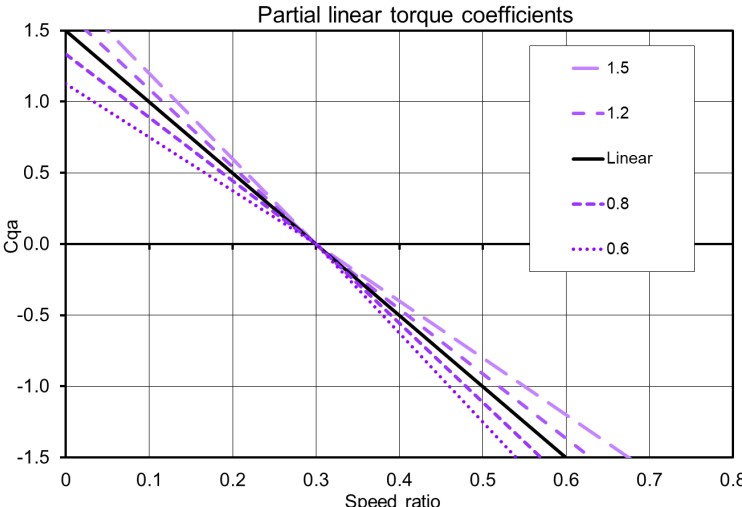

Figure 12  Partial linear torque coefficient curves for various κ ratios. Linear model in black

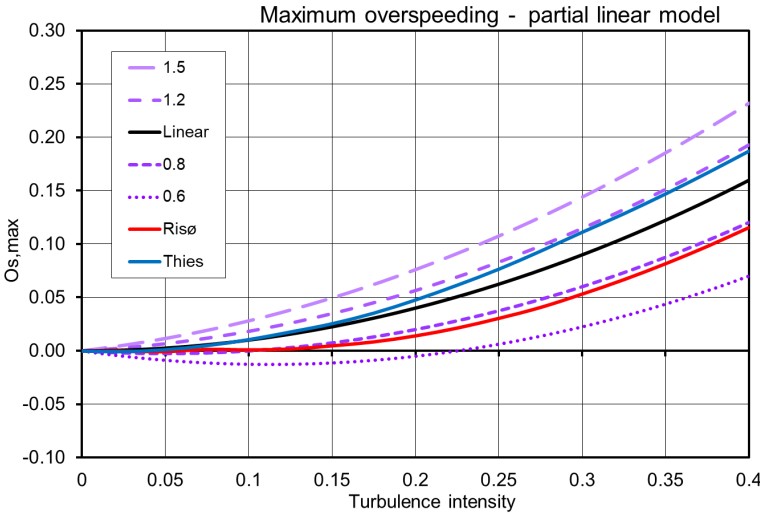

Figure 13  Maximum overspeeding of partial linear torque coefficients for various κ ratios

**6    Step responses derived from cup anemometer models**

**6.1    Step response with the parabolic torque coefficient model**

The differential equation for the parabolic torque coefficient model (7) is rearranged to an expression in $\omega$ (setting friction and threshold wind speed to zero):

$$\frac{d\omega}{dt} = \frac{\rho A R^3 \beta}{2I}(\omega - \frac{\lambda_0}{R}U)(\omega - \frac{\lambda_1}{R}U) \qquad (36)$$

We now make the substitution:





$$s = \frac{1}{\omega - \frac{\lambda_0}{R}U} \tag{37}$$

Which, expressed in rotor rotational speed is:

$$\omega = \frac{1}{s} + \frac{\lambda_0}{R}U \tag{38}$$

With the derivative:

$$d\omega = -\frac{1}{s^2}ds \tag{39}$$

Inserting expressions of the substitution, and rearranging, () becomes:

$$\frac{ds}{dt} + \frac{\rho AR^2\beta(\lambda_0 - \lambda_1)U}{2I}s = -\frac{\rho AR^3\beta}{2I} \tag{40}$$

Defining now the distance constant $l_0$, and inserting the slope of the torque coefficient curve $\kappa = \beta(\lambda_0 - \lambda_1)$ at the equilibrium speed ratio $\lambda_0$ we can express the distance constant as:

$$l_0 = -\frac{2I}{\rho AR^2\beta(\lambda_0 - \lambda_1)} = -\frac{2I}{\rho AR^2\kappa} \tag{41}$$

This distance constant is a general constant for a cup anemometer with a parabolic torque coefficient curve throughout the parabolic speed ratio range. Observe that the slope of the torque coefficient curve $\kappa$, at equilibrium speed ratio $\lambda_0$, is always negative, which makes the distance constant positive. Inserting and rearranging, the substituted differential equation is expressed in a simple and general form:

$$\frac{ds}{dt} - \frac{1}{l_0}Us = \frac{R}{l_0(\lambda_0 - \lambda_1)} \tag{42}$$

This equation is a first-order linear ordinary differential equation. It can be solved analytically for different input wind speeds

as function of time t. The general solution in s is:

$$s = \exp\left(-\frac{1}{l_0}\int_0^t U(t)dt\right)\left(-\frac{R}{l_0(\lambda_0 - \lambda_1)}\int_0^t \exp\left(\frac{1}{l_0}\int_0^t U(t)dt\right)dt + C\right) \tag{43}$$

Here C is a constant that must satisfy the starting requirements at $t = 0$. Inserting $s$ and rearranging we get the general analytical solution for the cup rotor angular speed for the parabolic torque coefficient model:

$$\omega = \frac{1}{\exp\left(-\frac{1}{l_0}\int_0^t U(t)dt\right)\left(-\frac{R}{l_0(\lambda_0 - \lambda_1)}\int_0^t \exp\left(\frac{1}{l_0}\int_0^t U(t)dt\right)dt + C\right)} + \frac{\lambda_0}{R}U(t) \tag{44}$$

If the cup anemometer is given a step input $\Delta U$ from $U_0$ to $U_s$, we find $C = -R/(\lambda_0\Delta U)$. Integrating and rearranging, we get the general solution to the step response of a cup anemometer with parabolic torque coefficient:

$$\omega = \frac{\lambda_0 U_s}{R}\left(1 - \frac{\exp\left(-\frac{U_s}{l_0}t\right)}{\frac{\lambda_0}{\lambda_0 - \lambda_1}\left(\exp\left(-\frac{U_s}{l_0}t\right) - 1\right) + \frac{U_s}{\Delta U}}\right) \tag{45}$$

For t going towards infinity the equation goes towards the static solution $\omega = \lambda_0 U_s/R$. For $\lambda_1 = 0$, the case with zero maximum overspeeding, we get the simpler equation:

$$\omega = \frac{\lambda_0 U_s}{R}\left(1 - \frac{\exp\left(-\frac{U_s}{l_0}t\right)}{\exp\left(-\frac{U_s}{l_0}t\right) - 1 + \frac{U_s}{\Delta U}}\right) \tag{46}$$

Figure 14 shows upwards step responses from 6.7m/s to 10m/s for the different torque coefficient curves in Fig. 10. Fig. 15 shows downwards step responses from 13.3m/s to 10m/s for the same torque coefficient curves. The corresponding speed ratio ranges are from 0.2 to 0.3 and from 0.4 to 0.3, so that we are within the speed ratio ranges where the torque coefficient curves



have negative slopes. The step responses deviate significantly. The torque coefficient curve with negative maximum overspeeding is the slowest in stepping up while it is the fastest in stepping down. The opposite is the case for the higher maximum overspeeding torque coefficients. Figure 16 shows the differences in stepping-up to stepping-down from Fig. 14 and Fig. 15. The very high and high overspeeding cases and the drag model case are speeding up faster than they speed down. The linear torque coefficient model have no difference between stepping up and stepping down, i.e. it speeds down just as fast

as it speeds up, but still it has a positive overspeeding with the turbulence intensity squared. The zero overspeeding case speeds down faster than it speeds up. This is a bit different than the commonly explained understanding that overspeeding is due to speeding up is faster than speeding down, (Busch, Kristensen 1976), (Wyngaard 1981), (IEA-11 1999). However, the simple explanation of the overspeeding concept is valid for varying wind (for example sinusoidal wind), and not for a constant wind, as in this case of step responses. Even though speeding down is equal to speeding up in step responses, there will still be an

overspeeding in a varying wind because the aerodynamic forces on the cup rotor are dependent on the wind speed squared. The torque coefficient curve has to counteract on this squared dependency to eliminate overspeeding, and the linear torque coefficient is not enough to do this. Only the parabolic torque coefficient curve with the second root through zero can achieve this requirement.

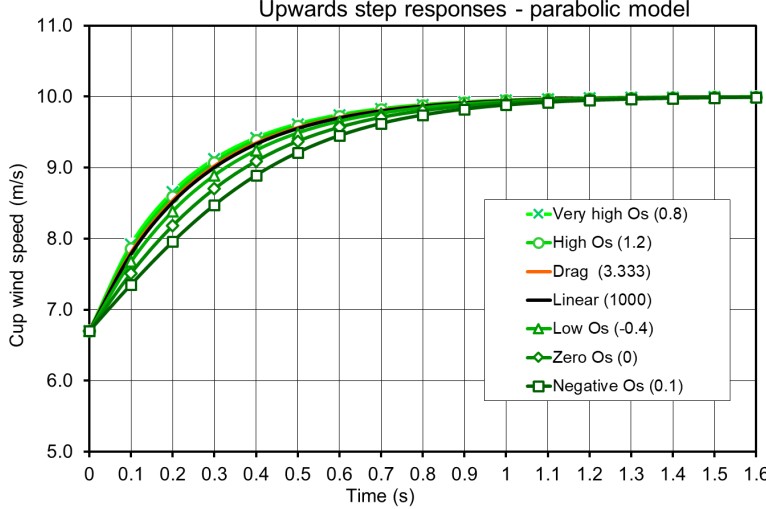

Figure 14  Step up response from 6.7m/s to 10m/s for cup anemometers with parabolic torque coefficient curves



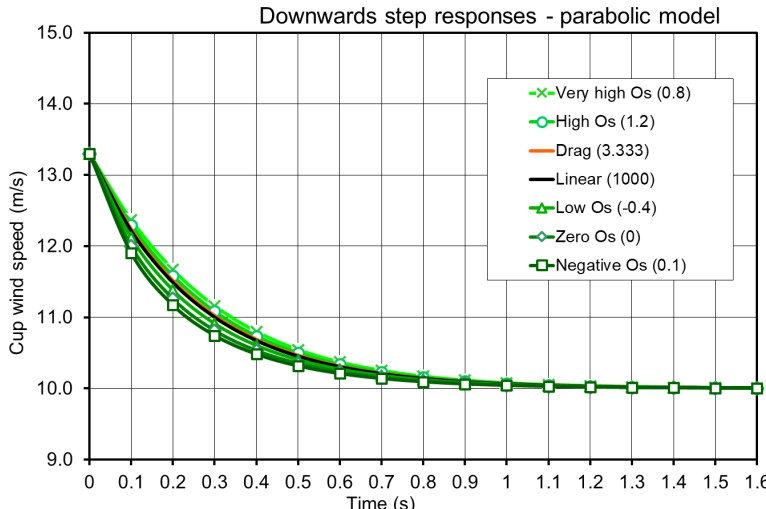

Figure 15  Step down response from 13.3m/s to 10m/s for cup anemometers with parabolic torque coefficient curve

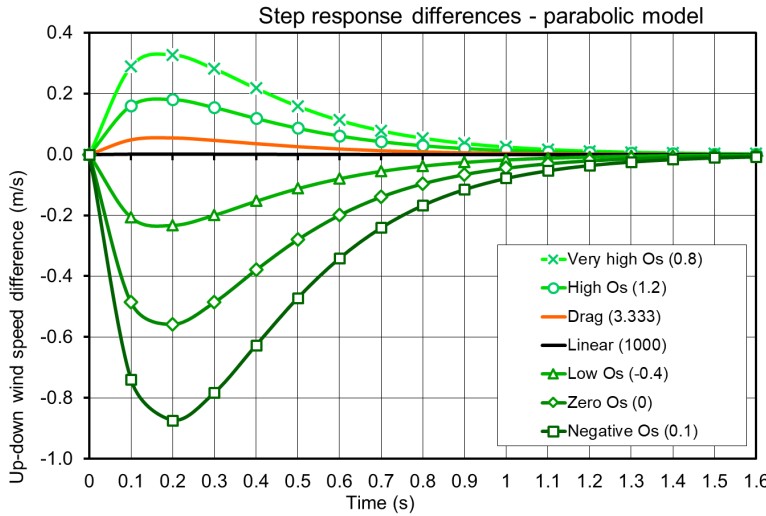

Figure 16  Step response differences between stepping up to 10m/s and stepping down to 10m/s for cup anemometers with
parabolic torque coefficient curves

### 6.2    Step response with the linear torque coefficient model

The linear torque coefficient model is expressed by:

$$C_{QA}(\lambda) = \frac{Q_A}{\frac{1}{2}\rho A R U^2} = \beta(\lambda - \lambda_0) = \kappa(\lambda - \lambda_0) \tag{47}$$

This expression can be interpreted as a special case of the parabolic torque coefficient model with $\lambda_1 \to \infty$ or $\lambda_1 \to -\infty$, as

shown before, and where the slope $\kappa$ is equal to the amplification factor $\beta$.

Linear model dynamics can by insertion of the distance constant $l_0$ be expressed by:

$$\frac{d\omega}{dt} + \frac{1}{l_0}U\omega = \frac{\lambda_0}{l_0 R}U^2 \tag{48}$$





This is a first order linear ordinary differential equation with the solution:

$$\omega = e^{-\int \frac{1}{l_0} U(t)dt} \left( \int \frac{\lambda_0}{l_0 R} U(t)^2 e^{\int \frac{1}{l_0} U(t)dt} dt + C \right) \tag{49}$$

With a step response from a wind speed $U_0$ and angular speed $\omega_0$ at time $t = 0$ to a tunnel wind speed $U_s = U_0 + \Delta U$ we find $C = \omega_0$ and:

$$\omega = \frac{\lambda_0}{R} \left( U_s - \Delta U \exp\left(-\frac{U_s}{l_0} t\right) \right) \tag{50}$$

This is the same expression we achieve for $\lambda_1 \to \infty$ or $\lambda_1 \to -\infty$ in equation () for the parabolic model. Inserting the time
constant $\tau = l_0/U_s$ we get:

$$\omega = \frac{\lambda_0}{R} \left( U_s - \Delta U \exp\left(-\frac{t}{\tau}\right) \right) \tag{51}$$

Equation () is equal to the step response formula in the IEA recommendation, (1999) for a step response from a certain rotor angular speed (over or under equilibrium speed ratio). Setting $\Delta U = U_s$ for a step response from stand still we get:

$$\omega = \frac{\lambda_0}{R} U_s \left(1 - \exp\left(-\frac{t}{\tau}\right)\right) \tag{52}$$

Equation () is equivalent to the step response equations from stand still described in the ISO standard, (ISO 2007), and the ASTM standard, (ASTM 2017). The IEA, ISO and ASTM documents describe methods to measure the distance constant with
step responses. They define the distance constant as the distance the air flows past a rotating anemometer during the time it takes the cup wheel to reach (1-1/e) or 63.2% of the equilibrium speed after a step change in wind speed. If we insert $t = \tau$ in equation () we find exactly this value. The IEA, ISO and ASTM documents, with their formulas, all relate to linear torque coefficient curves. Measuring the time to reach 63.2% of equilibrium speed corresponds to use torque coefficient data for speed ratios from zero to 0.632 times $\lambda_0$.

The IEA recommendation (1999) included a linear regression method for determination of the time constant $\tau$ in a step response. The time constant should be derived from equation (51) with a method to fit the data to the formula:

$$U = \omega \frac{R}{\lambda_0} = U_0 + \Delta U \left(1 - \exp\left(-\frac{t - t_0}{\tau}\right)\right) \tag{53}$$

Here $\tau = l_0/U_0$ where $l_0$ is the distance constant and $U_0$ is the constant wind speed during the step response. The method uses a linearization with the natural logarithm:

$$\log_n \left(1 - \frac{U - U_0}{\Delta U}\right) = -\frac{t - t_0}{\tau} \tag{54}$$

Pedersen (2011) used the IEA method but found the speed ratio ranges in the analysis (0% - 63.2%) being far from the range
that is most relevant. For the upwards step response, he found the appropriate equilibrium speed ratio range to be 50%-98%, and for the downwards step response 150%-102%. These speed ratio ranges would better represent the torque for the relevant turbulence intensities. The ISO method recommends 30-74%, but this range is also far from the relevant speed ratio range.

With a linear regression of the measured data in equation (54) the slope $-1/\tau$ of the step response may be determined, and from the slope $\tau$ is derived. With the distance constant relation $l_0 = \tau U_0$, the slope of the torque coefficient at $\lambda_0$ is found from
equation (41):

$$\kappa = -\frac{2I}{\rho A R^2 l_0} = -\frac{2I}{\rho A R^2 \tau U_0} \tag{55}$$

### 6.3    Step responses with the partial linear torque coefficient model

The partial linear torque coefficient model is of more interest than the linear model because torque coefficient curves of actual cup anemometers fit better to this model. For this model step responses can be used to determine the torque characteristics, as



shown in the former chapter. In this case we just have two different slopes to determine with step responses made from either

side of the equilibrium speed ratio. The early step response measurements by Barton (1989) actually found two different distance constants for each cup anemometer type, and these could have been used to determine partial linear torque coefficients. Step responses can be utilized in practice to determine the slopes $\kappa_{low}$ and $\kappa_{high}$, to fit to a partial linear torque model (Pedersen 2011). Methods to do this was adopted as an approximate method in the IEC standard (IEC-12-1 2017) as an alternative method, in case detailed and tabulated torque measurements are not available for classification.

**7   Distance constant**

In deriving the step response characteristics, the distance constant of a cup anemometer with a parabolic torque coefficient curve was defined as:

$$l_0 = -\frac{2I}{\rho AR^2\kappa} \tag{56}$$

This constant is a general constant within a parabolic torque coefficient speed ratio range, including the drag and linear models. We also found that the distance constant for step responses of cup anemometers in several standards and references is

determined from the step wind speed and the time constant:

$$l_0 = \tau U_0 \tag{57}$$

The deduction of a step response expression from a cup anemometer with a parabolic torque coefficient curve showed that these two expressions are coincident. The common assumptions and procedures must therefore be, that torque coefficient curves are parabolic. This is, however, an assumption far from correct, confirmed from Figure 5 and 6. And this is why distance constants derived with procedures from the standards ASTM (2017) and ISO (2007) may give quite different results

specifically between step responses from low and high speed ratios, but also between different wind speed step responses. From equation (56) it is seen that the distance constant is expressed directly as a function of the torque coefficient slope $\kappa$ at the equilibrium speed ratio $\lambda_0$. It makes much more sense to relate the distance constant to the tangent of the torque coefficient at equilibrium speed ratio, rather than to relate it to the time it takes the cup wheel to reach 63.2% of the equilibrium speed after a step change, as it is defined in the ASTM and ISO standards. Distance constants should be extracted from step response

data as close to equilibrium speed ratio as possible, as it is described in the procedure of IEC (IEC-12-1 2017) in order to make them relevant to wind speed measurements.

When Barton (1989) found different distance constants for a cup anemometer, it was a clear indication that torque curves did not follow parabolas. Barton found two distance constants of a cup anemometer, consistent with the theory of partial linear torque coefficient curves. The partial linear torque coefficient model is in many cases a better mathematical model than the

parabolic torque model in fitting torque data of modern cup anemometers with conical cups. But in fact, the distance constant is not an inherent constant of a cup anemometer, because the torque coefficient curve varies a lot more than a parabolic curve. For a detailed analysis, and specifically for an IEC classification, it is important to use the wind tunnel measured and tabulated torque coefficient curve.

**8   Optimized torque characteristics**

Dahlberg (2006) made a significant number of dynamic tests on cup anemometer configurations. He found that the overspeeding effect was primarily dependent on the cup rotor design, as shown with the Thies and Risø cup anemometers, Figure 6. It was a revelation that hemispherical cup rotors provide significantly more overspeeding than conical cups, the same as Scrase verified in 1944. Dahlberg found, however, that a fat cup anemometer body could spoil low overspeeding of a cup




rotor with conical cups. The findings indicate that cup anemometer overspeeding is dependent on the whole design of the
instrument. Good designs can almost eliminate the overspeeding effect while other designs trigger significant overspeeding.
For the cup anemometer rotor itself, the maximum overspeeding can be zero when the second root of a parabolic torque
coefficient curve is zero. An optimized cup anemometer rotor has to have good starting torque characteristics and this does
not imply zero torque at the second root. An optimized cup anemometer rotor should have an optimized parabolic torque
coefficient curve limited to an appropriate range around the equilibrium speed ratio $\lambda_0$, and with perhaps linear tangential
curves outside of this range. An optimized cup anemometer rotor with this type of torque coefficient curve would achieve zero
overspeeding for low and medium turbulence intensities, and increasing overspeeding for high turbulence intensities. The
requirement of low inertia of the cup rotor is well-known from research by meteorologists, but having part of the torque
coefficient curve with zero overspeeding is new.

Description of an optimized torque coefficient curve could start from rotor stand still. Schrenk (1929) estimated starting torque
from the drag model. He used the drag coefficients of hemispherical cups at straight angles to the wind, $C_{DH} = 1.33$ and $C_{DL} =$
0.33 to get the starting torque coefficient $C_{QA0} = C_{DH} - C_{DL} = 1.00$. Hoerner (1965) found a little higher values, $C_{DH} = 1.42$
and $C_{DL} = 0.38$. Brevort (1934) found $C_{DH} = 1.40$ and $C_{DL} = 0.40$ for a hemispherical cup and $C_{DH} = 1.40$ and $C_{DL} = 0.48$
for a conical cup. The Risø and Thies cup anemometers in Figure 5 seem to reach $C_{QA0} = 1.0$ at speed ratios about $\lambda = 0.1$.
where the curves are still going up. Torque coefficient measurements by Dahlberg (2006) support the limitation to $C_{QA0} = 1.0$.
Extrapolation of the Risø and Thies torque coefficient curves, however, reach 1.5 at $\lambda = 0$ for both, and we therefore set this
point as basis for extrapolation of the linear curves.

With these start up conditions, and with a determined equilibrium speed ratio $\lambda_0 = 0.3$ we let the linear torque coefficient
curve converge to the tangent of a zero overspeeding parabolic torque coefficient curve. The amplification factor $\beta$ of the zero
overspeeding parabolic torque coefficient curve is in this case:

$$\beta = -1.5/\lambda_2^2 \tag{58}$$

Here $\lambda_2$ is the speed ratio of the merging linear and parabolic model curves below $\lambda_0$.

The speed ratio variations of a cup anemometer are not symmetric around equilibrium speed ratio $\lambda_0$ in a natural varying wind.
For the maximum overspeeding cases with sinusoidal wind the speed ratio variations extend 1.4, 2.0 and 3.1 times higher to
high than to low speed ratio values for turbulence intensities 12%, 24% and 36%, respectively. High speed ratios are reached
when wind speed falls to lower values. In the limiting case at very high speed ratios the cup rotor runs in relatively calm wind,
and only the low drag of the cups produces torque. In this very high speed ratio case, the cup drag might be considered
proportional to the cup speed squared times three for the three cups. The optimum zero overspeeding speed ratio range is, in
the optimized case however, considered symmetric around the equilibrium speed ratio, although this is not optimum, but
perhaps more realistic. For higher speed ratios we assume a linear curve, tangent to the parabolic curve, until reaching the
limiting very high speed ratio case.

With this description of an optimized torque coefficient curve, and with the values $C_{QA0} = 1.0$, $C_{QA0lin} = 1.5$, $\lambda_0 = 0.3$, $\lambda_1 =$
0, and with intersection points between linear and parabolic curves $\lambda_2 = 0.28, 0.26$ and 0.24, respectively, three optimized
torque coefficient curves are shown in Fig. 17. The three $\lambda_2$ values correspond to 7%, 14% and 20% of equilibrium speed ratio,
respectively. The slope ratio $\kappa_{low}/\kappa_{high}$ for the linear parts corresponding to the three $\lambda_2$ values are 0.76, 0.58 and 0.43,
respectively. The Risø and Thies torque coefficients are shown in Fig. 17 as well. The Thies curve is seen to curve upwards
while the other curves are curving downwards. This indicates the tendency that torque coefficient curves need to have for more
optimum overspeeding characteristics.

The maximum overspeeding curves for torque alone, and otherwise with Risø cup anemometer dimensions and rotor inertia,
except for Thies, are calculated with the ACCUWIND code and are shown in Fig. 18. Included are also maximum overspeeding
curves for Risø and Thies.



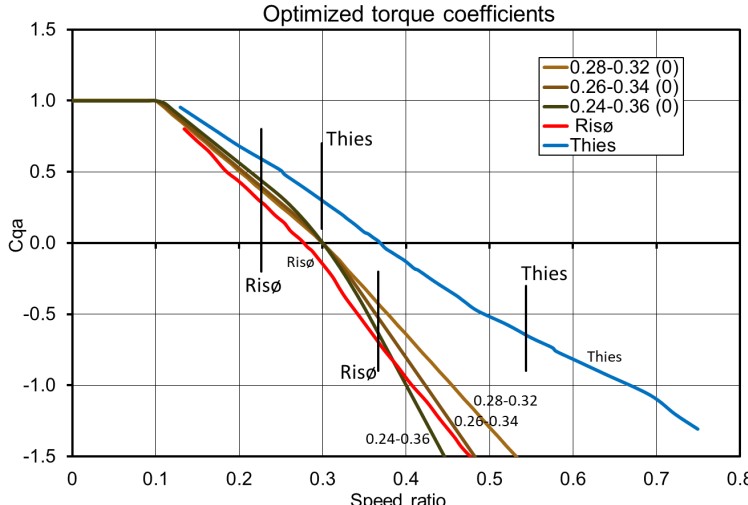


Figure 17  Optimized torque coefficient curves for $C_{QA0} = 1.5$, $\lambda_0 = 0.3$, $\lambda_1 = 0.0$. Parabolic ranges, 0.28-0.32 light pink, 0.26-0.34 medium pink, and 0.24-0.36 pink. Risø red, Thies blue, data from Figure 6. Vertical black lines are minimum and maximum speed ratio markings for 8m/s and 20% spectrum turbulence intensity.

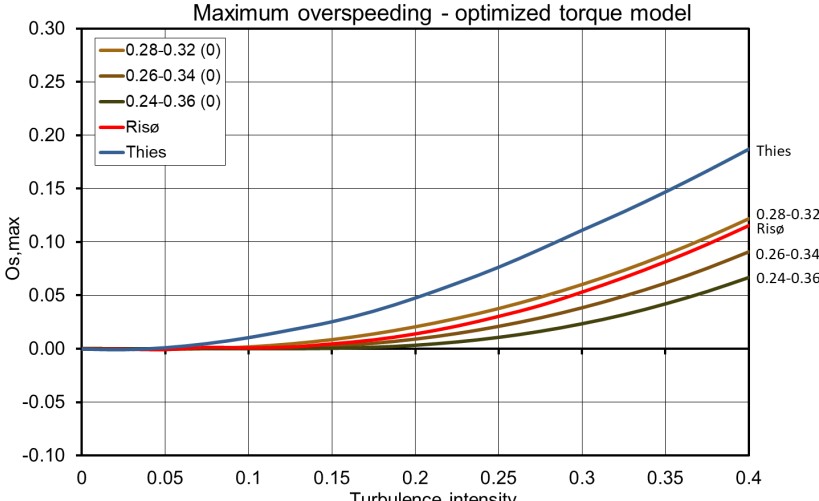

Figure 18 Maximum overspeeding of sinusoidal wind at 8m/s average wind speed. All calculations with Risø dimensions and inertia, except for Thies

The Thies cup anemometer seem to deviate from the optimized torque curves with significantly higher maximum overspeeding. The Risø cup anemometer seem to fit to the optimized torque curve shapes. A best fit might be to a 0.27-0.33 optimized torque curve.

The proposed optimum torque coefficient curves with zero overspeeding in certain speed ratio ranges have very low maximum overspeeding up to medium high turbulence intensity. For the Risø cup anemometer the low maximum overspeeding is up to about12% turbulence intensity. The Thies cup anemometer seems to do good up to about 5% turbulence intensity.

Low overspeeding in field measurements, requires low maximum overspeeding level as well as low inertia of the cup anemometer rotor. The maximum overspeeding level is independent of rotor inertia, but low rotor inertia can keep the cup

anemometer from operating too much time at the maximum overspeeding level.



The overspeeding in actual wind with a wind spectrum are calculated with the ACCUWIND model, using Mann turbulence code (1998) with a Kaimal spectrum and length scale $L_u = 350m$. Only torque is considered, i.e. no friction, tilt response is cosine shaped, and threshold wind speed set equal to zero. All calculations with Risø dimensions and rotor inertia, except for Thies, see Fig. 19.

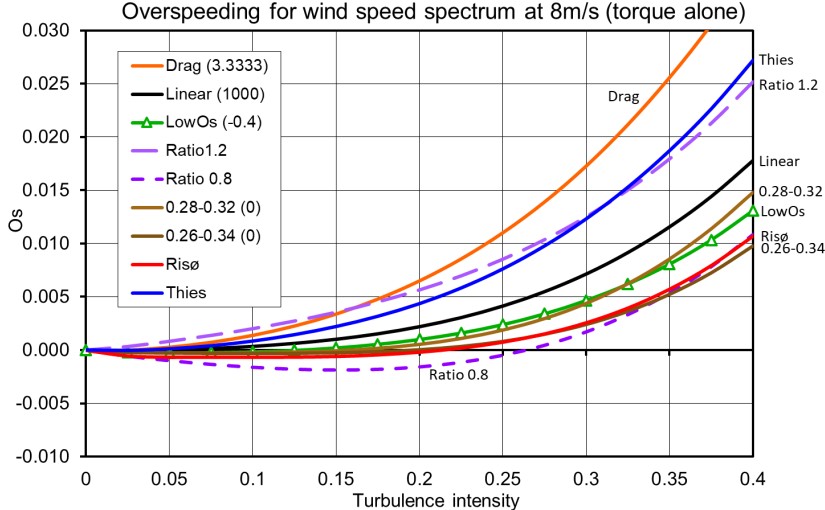


Figure 19 Overspeeding of Kaimal wind spectrum ($\sigma u/\sigma v/\sigma w=1/0.8/0.5$) at 8 m/s average wind speed. Curves include three parabolic model curves, two partial model curves, two optimized torque model curves, and Risø and Thies. All calculations with Risø dimensions and inertia, except for Thies

The wind spectra overspeeding show significantly reduced overspeeding from the maximum overspeeding curves. Remark

that the overspeeding scale is reduced by a factor ten. The Thies overspeeding is about one tenth of the maximum overspeeding and is about half between the linear and drag models, which are based on the Risø inertia. The Risø overspeeding is zero or slightly negative up to 20% turbulence and is reduced from 20% to 35% turbulence by about a factor twenty. Risø is very close to the 0.26-0.34 speed ratio case.

Calculations of the IEC classes A and B, where whole ranges of wind speed and turbulence spectra are included, are shown in

Table 1 for the torque characteristics alone. The two last rows includes also angular response and friction of Risø and Thies. The Thies classes 0.46A and 1.28B lies well between the linear and drag models, and Risø with classes 0.08A and 0.54B lies between the 0.28-0.32 and 0.26-0.34 optimum torque cases. The overspeeding values are in general not more than half a percent for class A and only the drag, ratio 1.2 and Thies cases come above 1% in class B. The classification changes significantly when the angular characteristics from Figure 3 and friction are included, and Thies classes changes to 1.55A and

7.23B, and Risø to 1.26A and 5.25B. The torque characteristics do only contribute with 30% for A and 18% for B for Thies, and 6% for A and 10% for B for Risø. Torque characteristics is not the main cause of systematic deviations. Angular characteristics take the lead here, but torque characteristics is still an important characteristic to take into account, especially when higher frequency content of wind spectra occur.

Table 1 IEC classification with the ACCUWIND model for torque alone (no friction, cos tilt response, zero threshold wind speed), except for last two rows for Risø and Thies, where all influence parameters are included. Torque curves from Figure 5, 10, 12 and 17. All calculations with Risø dimensions and inertia, except for Thies



| IEC classification/ACCUWIND model | IEC Class A<br>4-16 m/s<br>Ti $0.12 + 0.48/U_0$<br>Length scale $L_u$ 350 m<br>Air density<br>0.9-1.35 kg/m$^3$ | IEC Class B<br>4-16 m/s<br>Ti $0.12 + 0.48/U_0$<br>Length scale $L_u$ 350 m<br>Air density<br>0.9-1.35 kg/m$^3$ |
|---|---|---|
| Torque model | | |
| Linear (1000) | 0.24% | 0.88% |
| Drag (3.333) | 0.57% | 1.64% |
| LowOs (-0.4) | 0.15% | 0.66% |
| Ratio 1.2 | 0.49% | 1.27% |
| Ratio 0.8 | 0.21% | 0.51% |
| Range 0.28-0.32 (0) | 0.09% | 0.60% |
| Range 0.26-0.34 (0) | 0.07% | 0.49% |
| Range 0.24-0.36 (0) | 0.06% | 0.45% |
| Risø | 0.08% | 0.54% |
| Thies | 0.46% | 1.28% |
| Risø (all influence parameters) | 1.26% | 5.25% |
| Thies (all influence parameters) | 1.55% | 7.23% |

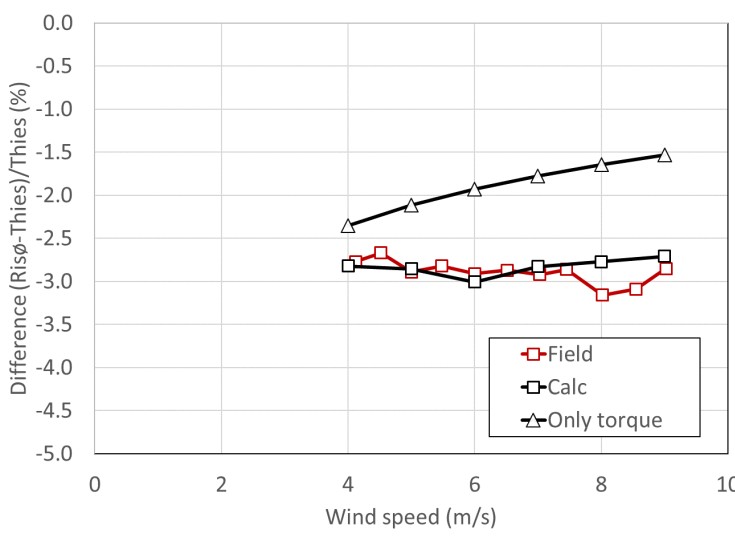

Figure 20 Differences between Thies and Risø cup anemometers in field comparison and ACCUWIND calculation, with all influence parameters and with only torque

The field comparison of the Thies and Risø cup anemometers in Fig. 1, which early demonstrated the problems of cup anemometer deviations, is in Fig. 20 supplemented with calculations. The calculations are made with a length scale $L_u$ =

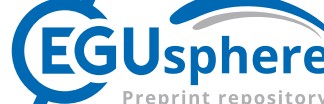

100m due to low height at 8 m, and with turbulences from 0.36 at 4 m/s to 0.31 at 8 m/s. More detailed knowledge of the field
conditions were not available. The contribution from torque characteristics is in this comparison significant due to the higher
frequency content in the wind spectrum, due to the low height.

In order to improve the torque characteristics to reduce the overspeeding effect we can start to look at the torque coefficient
curves of Risø and Thies from Fig. 5 in the most relevant speed ratio range, and normalize both curves with the speed ratio to
the equilibrium speed ratio 0.3, see Fig. 21. An improvement of the Risø torque characteristics could aim for the 0.26-0.34
curve, also shown in Fig. 21. We see that, below equilibrium speed ratio, where cup rotors accelerate, they almost fall on one
line with the same slope, except for the 0.24-0.36 curve. This part of the curves differ significantly from the parabolic model
curves in Fig. 10 and the partial linear model in Fig. 12, which spread quite a bit. The torque coefficient curves above the
equilibrium speed ratio, where cup rotors decelerate, spread significantly with steeper slopes for improved overspeeding. This
indicate that the overspeeding effect could be reduced by further increasing the low drag coefficient, as Brevort found when
going from a hemispherical cup ($C_{DL} = 0.40$) to a conical cup ($C_{DL} = 0.48$), while the high drag coefficient is the same for
both ($C_{DL} = 1.40$), (Brevort 1934). We cannot, however, use the drag model theory to improve on the overspeeding, though
the drag model is the only model which uses aerodynamic characteristics of the cup rotor in the torque coefficient expression,
$C_{DH}$ and $C_{DL}$. One could be tempted to increase the low drag coefficient further. The maximum overspeeding would be reduced
with the drag ratio according to equation (), but this would cause a reduction in sensitivity (calibration gain), equation (22) and
(10), which not is an advantage. To reduce the overspeeding effect it is necessary to consider the lift and drag interaction over
the whole revolution, including the flow in the 120° wake sector where one cup is in the wakes of the other two. Investigations
on such detailed complex flow in order to optimize torque characteristics has so far not been made.

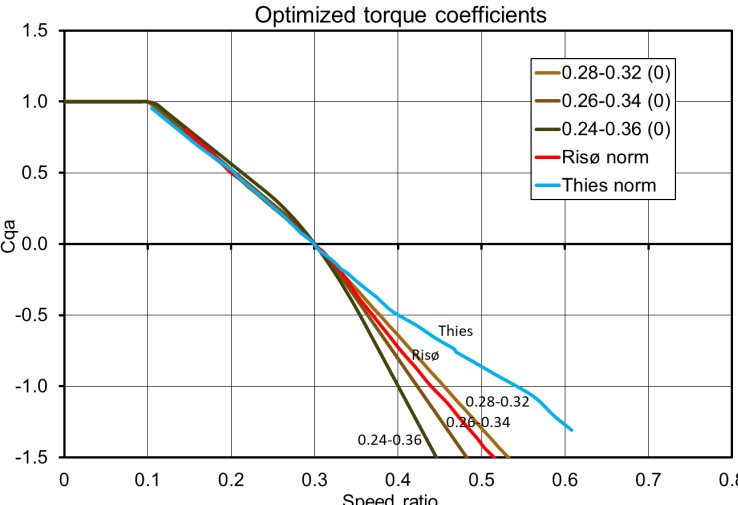

Figure 21 Speed ratio normalized torque coefficient curves for optimized torque with $C_{QA0} = 1.5$, $\lambda_0 = 0.3$, $\lambda_1 = 0.0$ and
three parabolic ranges and Risø and Thies

## 9   Conclusions

Within the last decades research on cup anemometer characteristics was taken to a new level within the wind energy
community. A historical review showed the need of improved models and methods for cup anemometer uncertainty analysis.
The development of improved cup anemometer models and classification methods of cup anemometers was triggered by the
measurement uncertainty requirements for power performance measurements on wind turbines. Results of the research are
now implemented in the IEC standards on power performance measurements, including the updated standards.



Inter-calibration of cup anemometers between European test stations revealed variations up to 10% between wind tunnel calibrations. However, European cooperation has today lead to variations below 0.5% within the MEASNET measurement organisation.

Assessments of cup anemometers by field comparisons showed variations of several percent, and the cause was found to be significantly dependent on turbulence intensity. European research projects SITEPARIDEN, CLASSCUP and ACCUWIND investigated the causes and found angular characteristics and dynamic response to be the main causes. Methods for assessment of characteristics, and models for systematic simulation of responses, and a classification method were developed. Cup anemometer models developed from research within the meteorology community were found, but no strict requirements or

methods for field measurement uncertainty was found.

The found cup anemometer models were the two-cup drag model, parabolic models, perturbation models, a phenomenological forcing model and linear models. All of the models were investigated and compared, in order to find an appropriate simulation model for uncertainty estimation and classification. None of the models fitted actual torque data accurate enough, and a new ACCUWIND model was developed, which use tabulated data instead of mathematical formulas.

Wind tunnel measurements analysed angular and dynamic response and found severe variations between commercial cup anemometer models. Dynamic response was investigated in wind tunnel with torque sensor measurements, and overspeeding was measured with sinusoidal varying wind in the wind tunnel. Very low overspeeding and even slightly negative overspeeding was experienced Maximum overspeeding as function of turbulence intensity, and step responses from below and above were found to express dynamic response in a clear way.

Comparison of models show that the often referenced drag model always lead to systematic high maximum overspeeding of about 1.1 times turbulence intensity squared, which however, is not present in modern cup anemometers with conical cups. The model fitted approximately to an older cup anemometer type with hemispherical cups. The more general parabolic model showed that maximum overspeeding can be zero and slightly negative at low and medium turbulence intensities. A new cup anemometer model with optimized zero maximum overspeeding was developed and a conical cup anemometer type was found

to fit approximately to the model. The linear torque coefficient model provides maximum overspeeding by the turbulence intensity squared. The partial linear model showed that torque characteristics can be measured with step responses from below and above. Such characteristics can approximately provide the same classification results as the ACCUWIND model with tabulated data.

When the models are exposed to a wind spectrum, the overspeeding is significantly reduced compared to the maximum

overspeeding. This is due to relatively low rotor inertia. The drag model shows the highest overspeeding, but the model is also significantly overestimating the overspeeding of the conical cup rotor.

Classification results with only torque characteristics (no friction, no angular response) show similar low overspeeding results. Classifications of the Risø and Thies cup anemometers show significantly higher values when angular response and friction are included.


**Code and data availability**

The ACCUWIND model and the classification method is described in detail in the IEC power performance standards. A model example calculation with an Anemcq7.exe code (available from the author) is provided in the standard.

**Author contributions**

TFP contributed with theoretical work, and he wrote the article. JÅD contributed with experimental wind tunnel work, with evaluation of results, discussions and review of the article.

**Competing interests**



TFP worked for decades on power performance measurements with the Risø cup anemometer, which was developed and improved at Risø (now DTU) by meteorologists, and later by Ole Frost Hansen, to whom the design and production was outsourced in 2006 in the spin-off company Windsensor. Both authors participated in the CLASSCUP and ACCUWIND projects and were members of the IEC standardisation committee that adopted the ACCUWIND classification method. Both authors declare to have no conflict of interest.


**Financial support**

The research work was supported by the two European projects CLASSCUP (contract JOR3-CT98-0263) and ACCUWIND (contract NNE5-2001-00831).

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
