# Peer review of "Modelling of cup anemometry and dynamic overspeeding in average wind speed measurements"

_EGUsphere, 2023_

## Author Comment (AC1)

Comments to: Modelling of cup anemometry and dynamic overspeeding in average wind speed measurements

Please be aware of the following editorial comments from Axel Albers, Windguard, Germany:

| Line | before | after |
|------|--------|-------|
| 9 | has | have |
| 217 | | defining I in text |
| 313 | Figure 6 | Figure 5 |
| 317 | Figure 6 | Figure 5 |
| 349 | equation() | equation(19) |
| 442 | () | equation(36) |
| 489 | equation () | equation(45) |
| 491 | equation() | equation(51) |
| 493 | equation() | equation(52) |
| 497 | equation() | equation(52) |
| 804 | " " | , , |

Best regards

Troels Friis Pedersen

---

## Author Response (AR2)

**Reply to reviewers on: Modelling of cup anemometry and dynamic overspeeding in average wind speed measurements**

**Reply to editorial comments from Axel Albers, Windguard, Germany through personal contact:**

| Line | before | after |
|------|--------|-------|
| 9 | has | have |
| 217 | | defining I in text |
| 313 | Figure 6 | Figure 5 |
| 317 | Figure 6 | Figure 5 |
| 349 | equation() | equation(19) |
| 442 | () | equation(36) |
| 489 | equation () | equation(45) |
| 491 | equation() | equation(51) |
| 493 | equation() | equation(52) |
| 497 | equation() | equation(52) |
| 804 | " " | , , |

**Reply to comments from Referee #1**

Ref#1: General comments:

The work is a significant analysis on different possible models to evaluate cup anemometers' overspeeding. The models are well explained and their performance when analyzing the maximum overspeeding or for a wind speed spectrum are correctly presented. Results, together with discussion about the step response and distance constant, provide helpful information for researchers who need to model cup anemometers.

Reply: Takes note of the positive comments

Ref#1: Specific comments:

In chapter 8, it becomes slightly confusing to have the data calculated with the RISØ cup anemometer dimensions and rotor inertia, but then have results also compared with Thies. A discussion about the implications of this fact in the analysis would be appropiate.

Reply: Admit that it might be sligtly confusing, and that a discussion on this will improve on understanding!  Adding sentence: Risø and Thies are actual cup anemometers with individual dimensions and rotor inertia, but the Risø cup anemometer is the one that is interesting to optimize incrementally, and why the Risø properties is used.

Ref#1: Technical corrections:

146: English correction. "could not be explained alone on distant constant values" (could not be explained by the distance constant values alone).

Reply: Corrected as proposed

Ref#1: Figure captions should state with more detail the anemometer models.

Reply:

Fig 6 added "Thies and Risø" cup anemometers

Fig 10 changed sentence from "Parabolic torque coefficient curves for …" to "Torque coefficient curves for parabolic model with equilibrium speed ratio $\lambda_0 = 0.3$, and slope at equilibrium speed ratio $\kappa = -5$. Various values of $\lambda_1$ as shown in the legend. Linear model in black and drag model in orange"

Fig 11 changed to "… Risø (red) and Thies (blue) are added", Fig 12 changed to "Torque coefficient curves for partial linear model with various $\kappa$ ratios. Linear model in black"

Fig 13 changed "coefficients" to "coefficient model"

Fig 17 changed from "…$\lambda_1 = 0.0$,  Parabolic ranges, 0.28-0.32 light pink, 0.26-0.34 medium pink, and 0.24-0.36 pink. Risø red, Thies blue, data from Fig. 6." to "…$\lambda_1 = 0.0$, and parabolic ranges, 0.28-0.32 (beige light), 0.26-0.34 (beige medium), and 0.24-0.36 (beige dark). Added Risø (red), Thies (blue), data from Fig. **Fejl! Henvisningskilde ikke fundet.**6."

Fig 18 changed to "Maximum overspeeding at sinusoidal wind of 8 m/s average wind speed for optimized torque coefficient curves from Figure 17, calculated with same dimensions and inertia as Risø. Added Risø (red) and Thies (blue, and with Thies properties)."

Fig 19 changed from "Overspeeding of Kaimal wind spectrum (σu/σv/σw=1/0.8/0.5) at 8 m/s average wind speed. Curves include three parabolic model curves, two partial model curves, two optimized torque model curves, and Risø and Thies. " to "Overspeeding for Kaimal wind spectrum (σu/σv/σw=1/0.8/0.5) at 8 m/s average wind speed. Curves include: three parabolic model curves, drag (orange), linear (black), low Os (green), two partial model curves, ratio 1.2 (long dashed purple), ratio 0.8 (short dashed purple), two optimized torque model curves, 0.28-0.32 (beige light), 0.26-0.34 (beige medium), and Risø (red) and Thies (blue)."

Fig 20 changed from "Differences between Thies and Risø cup anemometers in field comparison and ACCUWIND calculation, with all influence parameters and with only torque" to "Differences between Thies and Risø cup anemometers from the field comparison in Figure 1, and with two ACCUWIND calculations, one with all influence parameters, and one where only torque is considered."

Fig 21 changed from " Speed ratio normalized (to $\lambda_0 = 0.3$) torque coefficient curves for optimized torque with constants $C_{QA0} = 1.5$, $\lambda_0 = 0.3$, $\lambda_1 = 0.0$ and three parabolic ranges (0.24-0.36, 0.26-0.34 and 0.28-0.32), and Risø and Thies torque coefficient curves normalized as well" to "Torque coefficient curves for optimized torque with constants

$C_{QA0} = 1.5$, $\lambda_0 = 0.3$, $\lambda_1 = 0.0$ and three parabolic ranges 0.24-0.36 (light beige), 0.26-0.34 (medium beige), 0.28-0.32 (dark beige), and added Risø (red) and Thies (blue) torque coefficient curves, normalized to speed ratio $\lambda_0 = 0.3$"

Ref#1: 191: Reference missing about the Pedersen research

Reply: added ", (Dablberg et al. 2006),"

Ref#1: The paper needs a through correction on equation referencing (many equation references appear as ()). See 349 and 491 as examples.

Reply: Equation references are changed according to comments from Axel Albers:

| | | |
|---|---|---|
| 349 | equation() | to equation(19) |
| 442 | () | to equation(36) |
| 489 | equation () | to equation(45) |
| 491 | equation() | to equation(51) |
| 493 | equation() | to equation(52) |
| 497 | equation() | to equation(52) |

In line 657 the sentence with missing equation: "The maximum overspeeding would be reduced with the drag ratio according to equation (), but this would cause a reduction in sensitivity (calibration gain), equation (22) and (10), which not is an advantage." is changed to: "Increasing the low drag coefficient by 10% would increase the drag ratio k by 10%, and reduce the equilibrium speed ratio by 8%, equation (23). The calibration gain would be increased by 8%, equation (10), because the rotor would run slower, and the maximum overspeeding would be reduced by less than 2%, equation (23) and (24). The maximum overspeeding would for further increase of the low drag coefficient converge towards the linear maximum overspeeding, turbulence intensity squared, and the drag model cannot provide a lower value for any drag ratio.

Ref#1: 683: experienced. Maximum overspeeding (dot missing).

Reply: dot inserted

**Reply to comments from Referee #2:**

Ref#2: Excelent work on cup anemometer overspeeding.

Reply: Grateful and humble

Ref#2: Some minor comments:

 - Line 45. Sanz-Andres et al 2014. (Within References: Sanz-Andrés, A.; Pindado, S.; Sorribes, F. Mathematical analysis of the effect of the rotor geometry on cup

anemometer response. Sci. World J. 2014, Article ID 537813, DOI: 10.1155/2014/537813.)

- Line 679. "...mathematical formulae".

Reply: Forgot co-authors. This is corrected according to referee. Misspelling is corrected